# Policy Optimization via Adv2: Adversarial Learning on Advantage Functions

**Matthieu Jonckheere**  *matthieu.jonckheere@laas.fr*
*LAAS, Université de Toulouse, CNRS, Toulouse, France*

**Chiara Mignacco**  *chiara.mignacco@universite-paris-saclay.fr*
*Université Paris-Saclay, CNRS, Inria, Laboratoire de mathématiques d'Orsay, 91405, Orsay, France*

**Gilles Stoltz**  *gilles.stoltz@universite-paris-saclay.fr*
*Université Paris-Saclay, CNRS, Inria, Laboratoire de mathématiques d'Orsay, 91405, Orsay, France*

**Reviewed on OpenReview:** *https://openreview.net/forum?id=Oyueig1OEd*

## Abstract

We revisit the reduction of learning in adversarial Markov decision processes [MDPs] to adversarial learning based on $Q$–values; this reduction has been considered in a number of recent articles as one building block to perform policy optimization. Namely, we first consider and extend this reduction in an ideal setting where an oracle provides value functions: it may involve any adversarial learning strategy (not just exponential weights) and it may be based indifferently on $Q$–values or on advantage functions. We then present two extensions: on the one hand, convergence of the last iterate for a vast class of adversarial learning strategies (again, not just exponential weights), satisfying a property called monotonicity of weights; on the other hand, stronger regret criteria for learning in MDPs, inherited from the stronger regret criteria of adversarial learning called strongly adaptive regret and tracking regret. Third, we demonstrate how adversarial learning, also referred to as aggregation of experts, relates to aggregation (orchestration) of expert policies: we obtain stronger forms of performance guarantees in this setting than existing ones, via yet another, simple reduction. Finally, we discuss the impact of the reduction of learning in adversarial MDPs to adversarial learning in the practical scenarios where transition kernels are unknown and value functions must be learned. In particular, we review the literature and note that many strategies for policy optimization feature a policy-improvement step based on exponential weights with estimated $Q$–values. Our main message is that this step may be replaced by the application of any adversarial learning strategy on estimated $Q$–values or on estimated advantage functions. We leave the empirical evaluation of these twists for future research.

## 1 Introduction

In this article, we revisit a specific approach in policy optimization for adversarial Markov decision processes [MDPs] in the episodic setting, namely, the closed-form design of policies selected over time (which change incrementally) based on estimated value functions. In virtually all previous work, these policies are computed via the same adversarial-learning strategy, referred to under possibly different names: exponential weights, weighted majority, Boltzmann reweighting, or online mirror descent, to name a few. It turns out that different adversarial-learning strategies may be used, which may have important consequences in practice.

Put differently, this article aims to formally establish the mathematical consistency between the study of adversarial MDPs and (plain) adversarial learning, and to effectively bridge these two important areas of learning theory.

## 1.1 Brief literature review

Before reviewing in detail our contributions, we first provide a concise overview of the related literature and justify some claims contained in the previous paragraph.

**Adversarial MDPs / Reduction to adversarial learning.** The setting of adversarial MDPs was introduced by Even-Dar et al. (2009) and Yu et al. (2009). As in the standard episodic setup, the transition kernels dictating the evolution of the states are unknown and constant over time. However, the reward functions vary over time and may be chosen by some adversary; they are possibly revealed at the end of an episode. Both references were also the first ones to introduce a reduction of the control of adversarial MDPs to standard adversarial learning (a setting also called expert prediction; see Cesa-Bianchi & Lugosi, 2006 for an overview thereof). In this article, we will be interested in closed-form policy optimization, and not in approaches relying on so-called occupancy measures (introduced by Zimin & Neu, 2013), which solve a complex convex optimization problem at each episode, without resulting in closed-form expressions for the output policies (see, e.g., Rosenberg & Mansour, 2019).

**Policy optimization.** Policy optimization refers to designing policies to be used at each episode, often obtained by sequential incremental updates, and may be opposed to value-based learning in MDPs, which focuses on estimating and improving value functions rather than directly constructing policies. Several approaches were considered in policy optimization, for instance, (natural) policy gradient (Sutton et al., 2000, Kakade & Langford, 2002), and variants like Trust Region Policy Optimization or Proximal Policy Optimization (TRPO and PPO, respectively; see Schulman et al., 2015, Schulman et al., 2017). We will rather be interested in the closed-form policy design relying on estimates of $Q$–value functions. This vein of research includes the works by Shani et al. (2020), Cai et al. (2020), He et al. (2022), Zhao et al. (2023), Tiapkin et al. (2025) to name a few representative contributions (see also Abbasi-Yadkori et al., 2019). The settings differ in these articles depending, among others, on the feedback on the reward functions (full monitoring or bandit feedback) and on the structural assumptions, or lack thereof, on the transition kernels.

However, all cited references have one thing in common: they rely on the same adversarial-learning strategy to process the estimated $Q$–value functions (except Tiapkin et al., 2025, which builds on the present work).

**A single adversarial-learning strategy, based on exponential weights.** This same adversarial-learning strategy is known under different names and relies on exponential weights; for instance, Agarwal et al. (2021, Section 5.3) refers to it as multiplicative weights updates, Abbasi-Yadkori et al. (2019), as the Boltzmann policy[1], Shani et al. (2020) and Zhao et al. (2023), as online mirror descent (with a Kullback-Leibler regularization). Cai et al. (2020) and He et al. (2022) do not write any explicit strategy name for the corresponding step of their policy-optimization approach, but refer to the same closed-form update considered by earlier references; they obtain it by resorting to some follow-the-regularized-leader approach with an entropic regularization (known to lead to exponential weights, see Freund et al., 1997, Kivinen & Warmuth, 1999, Audibert, 2009).

Interestingly, this strategy based on exponential weights aligns with the concept of natural policy gradient for non-adversarial MDPs when the policy parametrization is softmax: both approaches involve the same update rule on the weights (this explicit update rule was, for instance, derived in Agarwal et al., 2021, Section 5.3, see also Kakade, 2001). This specific case, as the intersection of two optimisation paradigms, leads to remarkable theoretical guarantees in non-adversarial MDPs; see, in particular, the recent work by Müller & Montúfar (2024) and references therein.

Two exceptions to the use of the exponential-weight strategy are provided by Even-Dar et al. (2009) and Yu et al. (2009), which resort to a strategy called follow-the-perturbed-leader (Kalai & Vempala, 2005); but their setting and objectives are somewhat different from those considered in this article and the previous references.

---

[1]Abbasi-Yadkori et al. (2019) even states that "the choice of the Boltzmann policy is not arbitrary", but one of the points of the present article is to actually show the contrary: many other choices of adversarial-learning strategies are suitable.

**Previous reductions of learning in MDPs to adversarial learning.** We provide a specific analysis of the strategy based on exponential weights in Section 6, obtaining improved regret bounds compared to the analyses provided in the mentioned references. These analyses range from a few-line-long proof performing a direct reduction to adversarial learning in Shani et al. (2020) (a proof that we copy in Section 3.2 but that can be improved in the specific case of exponential weights), to longer proofs (possibly several pages, see, e.g., Zhao et al., 2023, Appendix A.1). The typical proofs are between these two extremes. In particular, to the best of our knowledge, no other proof than the one by Shani et al. (2020) clearly identifies a reduction, and all other proofs rather mimic and adapt[2] the analysis of exponential weights in adversarial learning, as in Agarwal et al. (2021, Section 5.3) or Cai et al. (2020). We note that the cited references actually run the exponential-weight strategy on estimated $Q$–values: more details are provided in Section 8.

In a nutshell, among all cited references, Shani et al. (2020) already clearly identified how to reduce learning in MDPs to adversarial learning, but only leveraged this fact for one specific adversarial learning strategy.

## 1.2 Contributions and outline of this article

In Section 2, we formally define the setting of episodic adversarial MDPs and state our objective: the minimization of a cumulative regret, defined as the sum of the differences between the value functions of the best stationary policy and of the output policies.

Section 3 recalls the reduction of learning in MDPs to adversarial learning as clearly stated in the course of a proof by Shani et al. (2020). We state the reduction in the ideal setting where an oracle provides, at the end of each episode, the value functions corresponding to the policy played—a restriction that we discuss and mitigate later in Section 8. We essentially replicate the proof by Shani et al. (2020), based on the performance difference lemma, and actually generalize it by considering a broad family of possible adversarial learning strategies, not just exponential weights with a constant learning rate. For instance, the `ML-Prod` and `ML-Poly` strategies (Gaillard et al., 2014, Gaillard et al., 2021) are suitable adversarial-learning strategies that exhibit in general much better empirical performance than exponential weights. Another observation is that the theoretical guarantees hold when adversarial-learning strategies are fed with advantage functions instead of $Q$–functions, which constitutes a second possible source of improved empirical performance.

We then discuss three extensions (convergence of the last iterate, stronger forms of regret, aggregation of policies) and present two twists: a special-case analysis for exponential weights with improved bounds, and how to use the general theory developed in practical scenarios where advantage functions must be estimated (in particular due to the transition kernels being unknown).

**Extension 1: convergence of the last iterate.** In the case where reward functions are constant over time, Section 4 focuses on a regret called simple regret, which measures the difference in performance between the best stationary policy and the last policy selected. Agarwal et al. (2021, Section 5.3) controlled this quantity for exponential weights with a constant learning rate (in the discounted setting). We show how to extend their argument to a large class of adversarial strategies satisfying a natural property that we call "monotonicity of weights".

**Extension 2: Stronger forms of regret.** Section 5 shows that the general reduction studied in Section 3 also works for a stronger notion of regret called strongly adaptive regret and consisting of studying the sums of differences in value functions over sub-intervals of time. As a consequence, the so-called tracking regret may also be controlled: therein, the comparison is made not to the best stationary policy, but to the best sequence of policies with few shifts. To the best of our knowledge, the control of such improved forms of regret for MDPs is an original contribution.

**The special case of exponential weights.** Section 6 leverages elements from Extensions 1 and 2 to show that when the adversarial learning strategy consists in using exponential weights with a constant learning

---

[2]Typical proofs usually consider the analysis of exponential weights based on telescoping Kullback-Leibler terms (as in Freund & Schapire, 1999), but we note that shorter analyses exist, e.g., based on Hoeffding's lemma (see Cesa-Bianchi & Lugosi, 2006, Section 2.2).

rate, the (cumulative) regret may be bounded by the number of shifts in the reward sequence. This provides yet another generalization of the results of Agarwal et al. (2021, Section 5.3). In addition, the proof technique by Agarwal et al. (2021, Section 5.3) seemed highly specific to the discounted setting: we provide instead a treatment for the episodic setting.

**Extension 3: Aggregation (orchestration) of expert policies.** Adversarial learning is sometimes called prediction with experts (see Cesa-Bianchi & Lugosi, 2006). Section 7 considers the case where policies selected over time are no longer learned in a direct tabular setting, but are obtained by (state-by-state and stage-by-stage) convex combinations of some expert policies. The aim is to mimic the performance of the overall best such convex combination. This methodology, which we call aggregation (or orchestration) of expert policies, is also referred to as learning from multiple oracles (which may be understood as a specific paradigm in the vast imitation-learning literature); see, for instance, Cheng et al., 2020 and Liu et al., 2023. We show that to address this problem, it suffices to consider expert policies as actions in a lifted MDP and apply all results described earlier in this article. We obtain stronger performance guarantees than in the cited references.

**Empirical impacts as future research directions.** Section 8 puts into perspective the design of policies studied in this article: in practice, advantage functions are unknown but may be estimated, so that the strategies studied earlier in this article should be run on these estimates. We review the literature of policy optimization to explain how and why the strategies proposed in the literature may be modified: their policy-improvement step, stated with exponential weights, may in fact rely on many other adversarial-learning strategies. This modification has no impact on the theoretical guarantees but could be impactful on the practical performance. We however leave the assessment of that potential practical impact for future research.

## 2 Setting and aims

**Notation.** We denote by $\mathcal{P}(\mathcal{X})$ the set of probability distributions over some set $\mathcal{X}$, either finite or given by an interval of $\mathbb{R}$ in the sequel. For an integer $n \geqslant 1$, let $[n] = \{1, \ldots, n\}$ denote the set of the first $n$ integers.

**Setting.** We consider an $H$–episodic and (obliviously) adversarial Markov decision process [MDP] with finite state and action spaces $\mathcal{S}$ and $\mathcal{A}$, of respective cardinalities $S$ and $A$: each episode $t \geqslant 1$ is of length $H \geqslant 1$ and is governed by transition kernels $\boldsymbol{\mathcal{T}} = (\mathcal{T}_h)_{h \in [H-1]}$, where $\mathcal{T}_h : \mathcal{S} \times \mathcal{A} \to \mathcal{P}(\mathcal{S})$, and by reward functions $\boldsymbol{\mathcal{R}}_t = (\mathcal{R}_{t,h})_{h \in [H]}$, where $\mathcal{R}_{t,h} : \mathcal{S} \times \mathcal{A} \to \mathcal{P}([0,1])$. The transition kernels are constant across episodes, while the reward functions $\boldsymbol{\mathcal{R}}_t$ vary between episodes; they may actually be picked by an adversary in an oblivious manner, i.e., the entire sequence $(\boldsymbol{\mathcal{R}}_t)_{t \geqslant 1}$ is determined by the adversary before the first episode takes place.

We denote by $r_{t,h} : \mathcal{S} \times \mathcal{A} \to [0,1]$ the mean-payoff function associated with $\mathcal{R}_{t,h}$, i.e., $r_{t,h}(s, a)$ is the expectation of the distribution $\mathcal{R}_{t,h}(s, a)$, for each $s \in \mathcal{S}$ and $a \in \mathcal{A}$.

A (stationary, or one-shot) policy $\boldsymbol{\pi} = (\pi_h)_{h \in [H]}$ is a sequence of mappings $\pi_h : \mathcal{S} \to \mathcal{P}(\mathcal{A})$; we denote by $\pi_h(\,\cdot\,|s)$ the probability distribution over actions that the policy uses in stage $h$ and state $s$. The learner should determine a policy $\boldsymbol{\pi}_t$ at the beginning of each episode $t \geqslant 1$, based on the information gained from rounds $\tau \leqslant t - 1$; that information includes at least the states observed and actions played therein, as well as the rewards obtained. In some scenarios, additional observations may be performed, which we will explicitly detail; for instance, the learning system may observe, among other things, the mean-payoff functions $\boldsymbol{r}_\tau = (r_{\tau,h})_{h \in [H]}$ at the end of episode $\tau$.

At the beginning of each episode $t \geqslant 1$, the same initial state $s_{t,1} = s_1$ is set. Then, at each stage $h \in [H-1]$, the learning system draws an action $a_{t,h} \sim \pi_{t,h}(\,\cdot\,|s_{t,h})$, after which it obtains and observes a stochastic reward drawn independently from $\mathcal{R}_{t,h}(s_{t,h}, a_{t,h})$, while the environment moves to a new state drawn as $s_{t,h+1} \sim \mathcal{T}_h(\,\cdot\,|s_{t,h}, a_{t,h})$. In the final stage, only an action $a_{t,H} \sim \pi_{t,H}(\,\cdot\,|s_{t,H})$ is drawn, and a reward drawn independently from $\mathcal{R}_{t,H}(s_{t,H}, a_{t,H})$ is obtained and observed. We do not introduce pieces of

---

**Box A: Policy optimization, for direct tabular learning**

**MDP parameters:** state space $\mathcal{S}$, action space $\mathcal{A}$, initial state $s_1 \in \mathcal{S}$, transition kernels $\mathcal{T}$

**Initialization:** The environment picks a sequence $(\mathcal{R}_t)_{t \geqslant 1}$ of reward functions

**For episodes** $t = 1, 2, \dots$**:**

1. The initial state is set to $s_{t,1} = s_1$

2. **For stages** $h = 1, \dots, H$**:**

   (a) The learner picks a policy $\pi_{t,h} : \mathcal{S} \to \mathcal{P}(\mathcal{A})$

   (b) and draws an action $a_{t,h} \sim \pi_{t,h}(\,\cdot\,|s_{t,h})$

3. The learner receives and observes a reward drawn independently from $\mathcal{R}_{t,h}(s_{t,h}, a_{t,h})$, with conditional expectation $r_{t,h}(s_{t,h}, a_{t,h})$

4. If $h \leqslant H - 1$, the next state $s_{t,h+1} \sim \mathcal{T}_h(\,\cdot\,|s_{t,h}, a_{t,h})$ is drawn

**Goal:** Minimize the regret $R_T = \max_{\boldsymbol{\pi}} \sum_{t=1}^{T} \left( V_1^{\boldsymbol{\pi}, \mathcal{R}_t}(s_1) - V_1^{\boldsymbol{\pi}_t, \mathcal{R}_t}(s_1) \right)$

---

notation for the rewards actually obtained as all arguments in this article will be based on value functions, which, by the tower rule, only depend on the mean-payoff functions.

More precisely, by the tower rule, the value function $V_h^{\boldsymbol{\pi}, \mathcal{R}_t}$ of a given stationary policy $\boldsymbol{\pi} = (\pi_j)_{j \in [H]}$ at episode $t \geqslant 1$ and started at stage $h \in [H]$ equals, for all $s \in \mathcal{S}$,

$$V_h^{\boldsymbol{\pi}, \mathcal{R}_t}(s) = \mathbb{E}^{\boldsymbol{\pi}, \mathcal{T}} \left[ \sum_{j=h}^{H} r_{t,j}(s_j, a_j) \,\middle|\, s_h = s \right], \tag{1}$$

where the piece of notation $\mathbb{E}^{\boldsymbol{\pi}, \mathcal{T}}$ indicates that actions $a_h$ and states $s_h$ in the expectation are governed by the policy $\boldsymbol{\pi}$ and the transition kernels $\mathcal{T}$, as described above.

## 2.1 First aim: direct tabular learning

We evaluate the policies $\boldsymbol{\pi}_t$ picked over time in terms of their value functions and are interested in mimicking the performance of the best stationary policy in hindsight. More precisely, the learning system aims to control

$$\forall T \geqslant 1, \qquad R_T = \max_{\boldsymbol{\pi}} \sum_{t=1}^{T} \left( V_1^{\boldsymbol{\pi}, \mathcal{R}_t}(s_1) - V_1^{\boldsymbol{\pi}_t, \mathcal{R}_t}(s_1) \right), \tag{2}$$

where the maximum is over all stationary policies $\boldsymbol{\pi}$. We write "$\forall T \geqslant 1$" to indicate that either the time horizon $T$ is unknown or the regret should be controlled for all time horizons. The regret $R_T$ involves a sum essentially because the reward functions $\mathcal{R}_t$ evolve over time in a possibly adversarial way; when they are constant over time, then convergence of the last iterate (i.e., of the $T$–th term in the sum above) may be achieved, see Section 4.

The aim described above is called direct tabular learning as policies $\boldsymbol{\pi}_t$ are picked by determining, for each stage $h$ and state $s$, the entire probability distribution $\pi_{t,h}(\,\cdot\,|s)$. The terminology is borrowed from Agarwal et al., 2021, Section 3.

The setting above is summarized in Box A.

**Alternative aim in Section 7.** The aim described in Box A may be difficult to complete when the number $A$ of actions is large. In addition, the learning system may sometimes have some prior information

given by a finite set of expert policies among which some policies could perform well (the subsets of these good-performing policies could possibly depend on the state). We therefore introduce an alternative aim in Section 7 called aggregation (or orchestration) of expert policies, but actually show that resolving this objective is equivalent in some sense to the aim described in Box A.

## 2.2 Additional notation

For later use, we define $Q$–values and advantage functions, and use the same notation as in (1) to that end. For any pair of stationary policy $\boldsymbol{\pi}$ and reward functions $\boldsymbol{\mathcal{R}}$, we define its $Q$–value function at episode $t \in [T]$, and started from stage $h \in [H]$, as

$$
Q_h^{\boldsymbol{\pi}, \boldsymbol{\mathcal{R}}_t} : (s, a) \in \mathcal{S} \times \mathcal{A} \longmapsto \mathbb{E}^{\boldsymbol{\pi}, \boldsymbol{\mathcal{T}}} \left[ \sum_{j=h}^{H} r_{t,j}(s_j, a_j) \ \middle| \ s_h = s, \ a_h = a \right],
$$

and its advantage function as

$$
A_h^{\boldsymbol{\pi}, \boldsymbol{\mathcal{R}}_t} : (s, a) \in \mathcal{S} \times \mathcal{A} \longmapsto Q_h^{\boldsymbol{\pi}, \boldsymbol{\mathcal{R}}_t}(s, a) - V_h^{\boldsymbol{\pi}, \boldsymbol{\mathcal{R}}_t}(s). \tag{3}
$$

We only keep in the notation $V_h$, $Q_h$, and $A_h$ the parameters $\boldsymbol{\pi}$ and $\boldsymbol{\mathcal{R}}_t$ that vary, and omit the transition kernels $\boldsymbol{\mathcal{T}}$. We use the short-hand notation

$$
A_h^{\boldsymbol{\pi}, \boldsymbol{\mathcal{R}}_t}(s, \cdot) = \left( A_h^{\boldsymbol{\pi}, \boldsymbol{\mathcal{R}}_t}(s, a) \right)_{a \in \mathcal{A}} \tag{4}
$$

to denote the vector of advantages of a stationary policy $\boldsymbol{\pi}$ for a given episode $t$ and a given stage $h$.

## 3 Methodology and core result: adversarial learning on advantage functions

**Contributions of this section.** *We recall how strategies designed to control the regret in the so-called adversarial setting, i.e., satisfying guarantees as described in Definition 1 below, may be used to construct policies so as to control the regret in terms of value functions. This observation was essentially already made in the literature, at least for exponential weights; see, for instance, how Shani et al. (2020, Section 6) handles the quantity called term (ii) in their proof.*

Before formally stating our main result, we briefly recall what the adversarial setting consists in, mostly to set our notation. We assume that the reader is familiar with the fundamental concepts and results of adversarial learning and refer to the monograph by Cesa-Bianchi & Lugosi (2006) for a more detailed exposition.

### 3.1 Reminder on adversarial learning

We provide a description where $K \geqslant 2$ refers to the number of options that the learning strategy has (the number of experts with the classic terminology of adversarial learning). In Section 3.2, we will identify this set $[K]$ of options with the set of actions $\mathcal{A}$.

At each round $t \geqslant 1$, based on the information collected during past rounds, a learning strategy picks a convex combination $w_t = (w_{t,1}, \dots, w_{t,K}) \in \mathcal{P}([K])$ while an opponent player simultaneously picks, possibly at random, a vector $g_t = (g_{t,1}, \dots, g_{t,K})$ of signed rewards. Both $w_t$ and $g_t$ are revealed at the end of the round. More formally, we mean that a learning strategy is a sequence $\varphi = (\varphi_t)_{t \geqslant 1}$ of functions $\varphi_t : \mathbb{R}^{K(t-1)} \to \mathcal{P}([K])$ and that $w_t = \varphi_t\big((g_\tau)_{\tau \leqslant t-1}\big)$ for $t \geqslant 1$. This formula means in particular that the initial vector $w_1 = \varphi_1(\emptyset)$ is constant.

**Definition 1** (adversarial-learning regret bound)**.** *A sequential strategy controls the regret in the adversarial setting with rewards bounded by $M > 0$ if there exists a sequence $(B_{T,K})_{T \geqslant 1}$ of positive numbers with $B_{T,K}/T \to 0$ and such that, against all opponent players sequentially picking reward vectors in $[-M, M]^K$,*

$$
\forall T \geqslant 1, \qquad \max_{k \in [K]} \sum_{t=1}^{T} g_{t,k} - \sum_{t=1}^{T} \sum_{j \in [K]} w_{t,j} \, g_{t,j} \leqslant 2M \, B_{T,K}.
$$

The optimal orders of magnitude of $B_{T,K}$ are $\sqrt{T \ln K}$ (see Cesa-Bianchi & Lugosi, 2006). In Definition 1, the strategy may know $M$ and rely on its value. On the contrary, the number $T$ of rounds is unknown and in fact, for the sake of exposition, Definition 1 requires a control of the adversarial regret for all $T \geqslant 1$, which imposes a mild restriction.

Two simple examples of strategies abiding by the constraints of Definition 1 are instances of the potential-based strategies by Cesa-Bianchi & Lugosi (2003). They are defined based on a sequence of non-decreasing functions $\Phi_t : \mathbb{R} \to [0, +\infty)$; they resort to $w_{1,k} = 1/K$ and

$$\forall t \geqslant 2, \qquad w_{t,k} = \frac{v_{t,k}}{\sum\limits_{j \in [K]} v_{t,j}}, \qquad \text{where} \qquad v_{t,k} = \Phi_t \left( \sum_{\tau=1}^{t-1} g_{\tau,k} - \sum_{\tau=1}^{t-1} \sum_{j \in [K]} w_{\tau,j} g_{\tau,j} \right). \tag{5}$$

**Example 1.** *Cesa-Bianchi & Lugosi (2003, Section 2) show that the strategy based on the constant polynomial potentials $\Phi_t \equiv \Phi : x \mapsto \left( \max\{x, 0\} \right)^{2 \ln K}$ provides the control $B_{T,K} = \sqrt{6T \ln K}$ for the regret in the adversarial setting.*

**Example 2.** *Auer et al. (2002) studied exponential potentials $\Phi_t(x) = \exp(\eta_t x)$ with time-varying learning rates $\eta_t = (1/M)\sqrt{(\ln K)/t}$. This sequential strategy controls the regret with $B_{T,K} = \sqrt{T \ln K}$ in the adversarial setting.*

A third example is of a different, not potential-based, nature.

**Example 3.** *The greedy projection algorithm of Zinkevich (2003) relies on a sequence $(\eta_t)_{t \geqslant 1}$ of positive step sizes and sets $w_{t+1} = \text{proj}(w_t + \eta_t g_t)$ for $t \geqslant 1$, where $w_1 = (1/K, \ldots, 1/K)$ and where $\text{proj}$ is the convex projection onto $\mathcal{P}([K])$ in the Euclidean norm. For the choices $\eta_t = (1/M)\sqrt{1/(2Kt)}$, this strategy controls the regret in the adversarial setting with $B_{T,K} = \sqrt{2KT}$.*

Dozens of strategies satisfying the guarantees of Definition 1 exist.

## 3.2 Policy optimization via adversarial learning on advantage functions

This section presents rather standard material and must be read accordingly. Indeed, what follows is a reduction that was essentially known, though it has previously been applied only with exponential weights and on $Q$–values rather than advantage functions. The proof follows the one by Shani et al. (2020, Section 6)—see also Agarwal et al. (2021, proof of Theorem 16)—, i.e., is based on the performance difference lemma.

We present the reduction in the ideal setting, where an oracle provides at the end of each episode $t$ the value functions of the policy $\boldsymbol{\pi}_t$ and of the reward function $\boldsymbol{\mathcal{R}}_t$ selected by the learning system and the environment, respectively. (The reward function $\boldsymbol{\mathcal{R}}_t$ does not need to directly be revealed, though, but only indirectly through the value functions.) We consider this ideal setting throughout this article, except in Section 8, where we explain how to leverage in practice the results developed in the ideal setting.

**Oracle 1.** *At the end of each episode $t \geqslant 1$, an oracle provides, for each $h \in [H]$, the value functions*

$$Q_h^{\boldsymbol{\pi}_t, \boldsymbol{\mathcal{R}}_t} : \mathcal{S} \times \mathcal{A} \to [0, H-h+1], \quad V_h^{\boldsymbol{\pi}_t, \boldsymbol{\mathcal{R}}_t} : \mathcal{S} \to [0, H-h+1], \quad A_h^{\boldsymbol{\pi}_t, \boldsymbol{\mathcal{R}}_t} : \mathcal{S} \times \mathcal{A} \to \left[ -(H-h+1), H-h+1 \right]$$

*of the policy $\boldsymbol{\pi}_t$ and of the reward function $\boldsymbol{\mathcal{R}}_t$ selected by the learning system and the environment, respectively.*

For each stage $h \in [H]$, we fix a sequential strategy $\varphi_h = (\varphi_{t,h})_{t \geqslant 1}$ in the adversarial setting, relying on reward vectors bounded by $M_h = H - h + 1$ and of dimension $K = A$, i.e., indexed by $\mathcal{A}$. We run these strategies on the advantage functions, in a stage-by-stage and state-by-state manner, as follows: for all $t \geqslant 1$,

$$\forall h \in [H], \quad \forall s \in \mathcal{S}, \qquad \pi_{t,h}(\cdot | s) = \varphi_{t,h} \left( \left( A_h^{\boldsymbol{\pi}_\tau, \boldsymbol{\mathcal{R}}_\tau}(s, \cdot) \right)_{\tau \leqslant t-1} \right), \tag{6}$$

where we used the notation defined in (4). We refer to this strategy as $(\varphi_h)_{h \in [H]}$–Adv2, for $(\varphi_h)_{h \in [H]}$–adversarial learning on advantage functions.

It constitutes a "theoretical" strategy, as it relies on the oracle knowledge of the advantage functions—an issue that we discuss and mitigate later in Section 8. The strategy could be run instead on $Q$–values, see Remark 1 below.

**Theorem 1.** *In the setting of Section 2 where rewards lie in $[0, 1]$, if, for all $h \in [H]$, the sequential strategies $\varphi_h$ control the regret in the adversarial setting (Definition 1) by $B_{T,A}$ for $A$–dimensional reward vectors bounded by $H - h + 1$, then the $(\varphi_h)_{h \in [H]}$–Adv2 strategy defined in (6) controls the regret as:*

$$\forall T \geqslant 1, \qquad \max_{\boldsymbol{\pi}} \sum_{t=1}^{T} \left( V_1^{\boldsymbol{\pi}, \boldsymbol{\mathcal{R}}_t}(s_1) - V_1^{\boldsymbol{\pi}_t, \boldsymbol{\mathcal{R}}_t}(s_1) \right) \leqslant H(H+1) B_{T,A} \,.$$

As indicated above, following Shani et al. (2020, Section 6), the (short) proof of Theorem 1 relies on the so-called performance difference lemma, which we recall next. For the sake of completeness, references for this lemma and a proof thereof are provided in Appendix B.

**Lemma 1** (Performance difference lemma). *Let $\mu_{h'}^{s_1, \boldsymbol{\pi}, \boldsymbol{\mathcal{T}}}$ be the distribution of the state $s_{h'}$ of the $h'$–th stage, starting from the state $s_1$ in the first stage, following the stationary policy $\boldsymbol{\pi}$ and the transition kernels $\boldsymbol{\mathcal{T}}$. In a MDP with transition kernels $\boldsymbol{\mathcal{T}}$, for all pairs $\boldsymbol{\pi}, \boldsymbol{\pi}'$ of stationary policies, for all reward functions $\boldsymbol{\mathcal{R}}$, and for all stages $h \in [H]$,*

$$\sum_{s \in \mathcal{S}} \mu_h^{s_1, \boldsymbol{\pi}, \boldsymbol{\mathcal{T}}}(s) \left( V_h^{\boldsymbol{\pi}, \boldsymbol{\mathcal{R}}}(s) - V_h^{\boldsymbol{\pi}', \boldsymbol{\mathcal{R}}}(s) \right) = \sum_{h'=h}^{H} \sum_{s \in \mathcal{S}} \mu_{h'}^{s_1, \boldsymbol{\pi}, \boldsymbol{\mathcal{T}}}(s) \sum_{a \in \mathcal{A}} \pi_{h'}(a|s) A_{h'}^{\boldsymbol{\pi}', \boldsymbol{\mathcal{R}}}(s, a) \,.$$

*In particular, for $h = 1$,*

$$V_1^{\boldsymbol{\pi}, \boldsymbol{\mathcal{R}}}(s_1) - V_1^{\boldsymbol{\pi}', \boldsymbol{\mathcal{R}}}(s_1) = \sum_{h'=1}^{H} \sum_{s \in \mathcal{S}} \mu_{h'}^{s_1, \boldsymbol{\pi}, \boldsymbol{\mathcal{T}}}(s) \sum_{a \in \mathcal{A}} \pi_{h'}(a|s) A_{h'}^{\boldsymbol{\pi}', \boldsymbol{\mathcal{R}}}(s, a) \,.$$

*Proof of Theorem 1.* We fix a stationary policy $\boldsymbol{\pi}$ throughout the proof and control the regret with respect to this $\boldsymbol{\pi}$.

The first part consists of applying the adversarial-learning regret upper bound for each $h \in [H]$. As the rewards take values in $[0, 1]$, we have that $\left| A_h^{\boldsymbol{\pi}_\tau, \boldsymbol{\mathcal{R}}_\tau}(s, a) \right| \leqslant H - h + 1$ for all $\tau, s, a$. By the definition of advantage functions (for the equality to 0) and by Definition 1 and the design of the $(\varphi_h)_{h \in [H]}$–Adv2 strategy (for the upper bound), we have, for all $s \in \mathcal{S}$,

$$\max_{a \in \mathcal{A}} \sum_{t=1}^{T} A_h^{\boldsymbol{\pi}_t, \boldsymbol{\mathcal{R}}_t}(s, a) - \overbrace{\sum_{t=1}^{T} \sum_{a \in \mathcal{A}} \pi_{t,h}(a|s) A_h^{\boldsymbol{\pi}_t, \boldsymbol{\mathcal{R}}_t}(s, a)}^{=0} \leqslant 2(H - h + 1) B_{T,A} \,. \tag{7}$$

The second part consists of applying the performance difference lemma, i.e., Lemma 1 above with $h = 1$, which guarantees that

$$V_1^{\boldsymbol{\pi}, \boldsymbol{\mathcal{R}}_t}(s_1) - V_1^{\boldsymbol{\pi}_t, \boldsymbol{\mathcal{R}}_t}(s_1) = \sum_{h=1}^{H} \sum_{s \in \mathcal{S}} \mu_h^{s_1, \boldsymbol{\pi}, \boldsymbol{\mathcal{T}}}(s) \sum_{a \in \mathcal{A}} \pi_h(a|s) A_h^{\boldsymbol{\pi}_t, \boldsymbol{\mathcal{R}}_t}(s, a) \,.$$

Summing this equality over $t$ and rearranging, we get

$$\sum_{t=1}^{T} \left( V_1^{\boldsymbol{\pi}, \boldsymbol{\mathcal{R}}_t}(s_1) - V_1^{\boldsymbol{\pi}_t, \boldsymbol{\mathcal{R}}_t}(s_1) \right) = \sum_{h=1}^{H} \sum_{s \in \mathcal{S}} \mu_h^{s_1, \boldsymbol{\pi}, \boldsymbol{\mathcal{T}}}(s) \sum_{a \in \mathcal{A}} \pi_h(a|s) \sum_{t=1}^{T} A_h^{\boldsymbol{\pi}_t, \boldsymbol{\mathcal{R}}_t}(s, a)$$

$$\leqslant \sum_{h=1}^{H} \sum_{s \in \mathcal{S}} \mu_h^{s_1, \boldsymbol{\pi}, \boldsymbol{\mathcal{T}}}(s) \underbrace{\max_{a \in \mathcal{A}} \sum_{t=1}^{T} A_h^{\boldsymbol{\pi}_t, \boldsymbol{\mathcal{R}}_t}(s, a)}_{\leqslant 2(H-h+1) B_{T,A}} \leqslant 2 \underbrace{\sum_{h=1}^{H} (H - h + 1)}_{= H(H+1)} B_{T,A} \,, \tag{8}$$

where we substituted (7). Here, we crucially used that the weights $\mu_h^{s_1, \boldsymbol{\pi}, \boldsymbol{\mathcal{T}}}(s)$ are independent of $t$ as they only depend on the fixed benchmark policy $\boldsymbol{\pi}$, on the common transition kernels $\boldsymbol{\mathcal{T}}$, and on the initial state $s_1$ (identical for all $t$). $\qquad \square$

### 3.3 Comments

In this section, we comment and discuss the `Adv2` strategy (6) and its bound.

We first note that the regret bound of Theorem 1 is independent of the size $S$ of the state space; it only depends on the size $A$ of the action space, on the number $T$ of episodes, and on the length $H$ of the episodes. Given that adversarial-learning strategies have a per-round computational complexity typically proportional to $K$ (with the notation of Section 3.1), the per-round computational complexity of the `Adv2` strategies (6) are typically proportional to $SAH$ as far as the weight updates are concerned. The main computational issue lies in computing (or estimating, see Section 8) the advantage functions $A_h^{\boldsymbol{\pi}_\tau, \boldsymbol{\mathcal{R}}_\tau}$.

Second, for potential-based strategies (5), we note that the original definition (6) of `Adv2` and the alternative definition based on $Q$–values,

$$\pi_{t,h}(\,\cdot\,|s) = \varphi_{t,h}\Big(\big(Q_h^{\boldsymbol{\pi}_\tau, \boldsymbol{\mathcal{R}}_\tau}(s,\,\cdot\,)\big)_{\tau \leqslant t-1}\Big), \tag{9}$$

lead to the exact same strategies. This may be shown by induction, based on the fact that for all $h \in [H]$ and $(s,a) \in \mathcal{S} \times \mathcal{A}$, as in (7) for the first equality and due to the definitions of value functions for the second equality,

$$\sum_{\tau=1}^{t-1} A_h^{\boldsymbol{\pi}_\tau, \boldsymbol{\mathcal{R}}_\tau}(s,a) - \overbrace{\sum_{\tau=1}^{t-1} \sum_{a \in \mathcal{A}} \pi_{\tau,h}(a|s)\, A_h^{\boldsymbol{\pi}_\tau, \boldsymbol{\mathcal{R}}_\tau}(s,a)}^{=0} = \sum_{\tau=1}^{t-1} A_h^{\boldsymbol{\pi}_\tau, \boldsymbol{\mathcal{R}}_\tau}(s,a)$$

$$\text{and} \quad \sum_{\tau=1}^{t-1} Q_h^{\boldsymbol{\pi}_\tau, \boldsymbol{\mathcal{R}}_\tau}(s,a) - \underbrace{\sum_{\tau=1}^{t-1} \sum_{a \in \mathcal{A}} \pi_{\tau,h}(a|s)\, Q_h^{\boldsymbol{\pi}_\tau, \boldsymbol{\mathcal{R}}_\tau}(s,a)}_{=V_h^{\boldsymbol{\pi}_\tau, \boldsymbol{\mathcal{R}}_\tau}(s)} = \sum_{\tau=1}^{t-1} A_h^{\boldsymbol{\pi}_\tau, \boldsymbol{\mathcal{R}}_\tau}(s,a)\,.$$

For general adversarial-learning strategies, the induced strategies (6) and (9) may differ, though they achieve the same regret guarantees, as detailed by the following remark.

**Remark 1.** *An inspection of the proof of Theorem 1 shows that it would also work for the strategies of the form* (9). *Indeed, the inequality* (7) *therein would be replaced equivalently by*

$$2(H - h + 1)\, B_{T,A} \geqslant \max_{a \in \mathcal{A}} \sum_{t=1}^{T} Q_h^{\boldsymbol{\pi}_t, \boldsymbol{\mathcal{R}}_t}(s,a) - \overbrace{\sum_{t=1}^{T} \sum_{a \in \mathcal{A}} \pi_{t,h}(a|s)\, Q_h^{\boldsymbol{\pi}_t, \boldsymbol{\mathcal{R}}_t}(s,a)}^{=V_h^{\boldsymbol{\pi}_t, \boldsymbol{\mathcal{R}}_t}(s)} = \max_{a \in \mathcal{A}} \sum_{t=1}^{T} A_h^{\boldsymbol{\pi}_t, \boldsymbol{\mathcal{R}}_t}(s,a)\,,$$

*while the rest of the proof would be unaffected. However, using the advantage functions is preferred in practice, as it provides a greater numerical stability, as well as a possibly lower variance when the value function are estimated (see Section 8).*

## 4  Extension 1: Convergence of the last iterate for some adversarial learning strategies

**Contributions of this section.**  *We generalize an argument of Agarwal et al. (2021, Section 5.3), which was provided for exponential weights only (in the discounted setting): the aim is to control the convergence of the last iterate, i.e., to upper bound* $\max_{\boldsymbol{\pi}} V_1^{\boldsymbol{\pi}, \boldsymbol{\mathcal{R}}}(s_1) - V_1^{\boldsymbol{\pi}_T, \boldsymbol{\mathcal{R}}}(s_1)$, *when (mean) rewards functions are constant over time.*

*To do so, we introduce a concept of independent interest: monotonicity of weights for adversarial-learning strategies.*

More precisely, for adversarial-learning strategies $\varphi$ satisfying this property of monotonicity of weights, and in case reward functions do not vary over time (or even just mean reward functions do not vary over time, see Remark 2) the result of Theorem 1 may be strengthened into a convergence result of the last iterate, at a rate faster by a factor of $1/T$ compared to the convergence of the cumulative regret (2).

**Definition 2** (monotonicity of weights). *A sequential strategy $\varphi = (\varphi_t)_{t \geqslant 1}$ in the adversarial setting satisfies monotonicity of weights if against all opponent players sequentially picking $K$–dimensional reward vectors $g_\tau = (g_{\tau,k})_{k \in [K]}$, the convex weights output by $\varphi$ are such that*

$$\forall t \geqslant 1, \qquad \sum_{k \in [K]} w_{t+1,k} \left( g_{t,k} - \sum_{j \in [K]} w_{t,j} g_{t,j} \right) \geqslant 0 \,,$$

*where we recall the notation $(w_{t,k})_{k \in [K]} = \varphi_t(g_1, \ldots, g_{t-1})$ and $(w_{t+1,k})_{k \in [K]} = \varphi_t(g_1, \ldots, g_{t-1}, g_t)$.*

The proof of Lemma 2 below explains why the property of Definition 2 is termed monotonicity of weights, and why it is a natural property of an adversarial learning strategy: indeed, the property is satisfied as soon as weights for components $k$ associated with a good (respectively, bad) reward $g_{t,k}$ in the previous round increase (respectively, decrease), where good or bad is determined by the sign of what is called the instantaneous regret with respect to component $k$ in round $t$:

$$g_{t,k} - \sum_{j \in [K]} w_{t,j} g_{t,j} \,.$$

**Lemma 2.** *The potential-based strategies (5) of Cesa-Bianchi & Lugosi (2003) with constant, non-decreasing potential functions $\Phi_t \equiv \Phi$ (like in Example 1) and the greedy projection algorithm (Example 3) of Zinkevich (2003) satisfy monotonicity of weights.*

*Proof.* We start with the potential-based strategies (5), in case of a constant, non-decreasing potential function $\Phi_t \equiv \Phi$, and use the notation defined therein. For each $t \geqslant 1$, since $\Phi$ is non-decreasing, we have, for all $k \in [K]$,

$$v_{t+1,k} \geqslant v_{t,k} \quad \Longleftrightarrow \quad g_{t,k} - \sum_{j \in [K]} w_{t,j} g_{t,j} \geqslant 0 \,, \qquad \text{thus} \qquad (v_{t+1,k} - v_{t,k}) \left( g_{t,k} - \sum_{j \in [K]} w_{t,j} g_{t,j} \right) \geqslant 0$$

in all cases. Therefore,

$$\sum_{k \in [K]} v_{t+1,k} \left( g_{t,k} - \sum_{j \in [K]} w_{t,j} g_{t,j} \right) \geqslant \sum_{k \in [K]} v_{t,k} \left( g_{t,k} - \sum_{j \in [K]} w_{t,j} g_{t,j} \right) = 0 \,,$$

where the equality to 0 and the final result of Definition 2 are obtained, respectively, by normalizing the $v_{t+1,k}$ and $v_{t,k}$ into $w_{t+1,k}$ and $w_{t,k}$.

For the greedy projection algorithm (Example 3) of Zinkevich (2003), we note that by a property of Euclidean projections onto a convex set (here, $w_{t+1}$ is the projection of $w_t + \eta_t\, g_t$ onto the simplex, and $w_t$ also belongs to the simplex), the following Euclidean inner product is non-positive:

$$0 \geqslant \langle w_t - w_{t+1}, \, (w_t + \eta_t\, g_t) - w_{t+1} \rangle = \|w_t - w_{t+1}\|^2 + \eta_t \langle w_t - w_{t+1}, \, g_t \rangle \,,$$

so that $\langle w_{t+1} - w_t, \, g_t \rangle \geqslant 0$, which is exactly monotonicity of weights. $\qquad\square$

We are now ready to state our result of convergence of the last iterate, which generalizes an argument of Agarwal et al. (2021, Section 5.3).

**Theorem 2.** *Assume reward functions do not vary over time and are all equal to some $\mathcal{R}$. If, for all $h \in [H]$, the sequential strategies $\varphi_h$ satisfy monotonicity of weights (Definition 2) and control the regret in the adversarial setting (Definition 1) by $B_{T,A}$ for $A$–dimensional reward vectors bounded by $H - h + 1$, then the last iterate of the $(\varphi_h)_{h \in [H]}$–$\mathtt{Adv2}$ strategy defined in (6) satisfies*

$$\forall T \geqslant 1, \qquad \max_{\boldsymbol{\pi}} V_1^{\boldsymbol{\pi},\mathcal{R}}(s_1) - V_1^{\boldsymbol{\pi}_T,\mathcal{R}}(s_1) \leqslant \frac{H(H+1)\, B_{T,A}}{T} \,.$$

The bound by Agarwal et al. (2021, Section 5.3), where the exponential weights with a constant learning rate are considered, corresponds to this theorem but is stated separately in Corollary 2, for reasons that will be made clear in Section 6. As the proof of Theorem 2 is concise, we provide it in the main body of this article.

*Proof.* Given the definition (6), the monotonicity of weights (Definition 2), and the definition of advantage functions, we have that, for all $t \geqslant 1$, for all $h \in [H]$, and $s \in \mathcal{S}$,

$$\sum_{a \in \mathcal{A}} \pi_{t+1,h}(a|s) A_h^{\boldsymbol{\pi}_t, \boldsymbol{\mathcal{R}}}(s, a) \geqslant \sum_{a \in \mathcal{A}} \pi_{t,h}(a|s) A_h^{\boldsymbol{\pi}_t, \boldsymbol{\mathcal{R}}}(s, a) = 0 \,.$$

Therefore, the performance difference lemma, i.e., Lemma 1 above with $h = 1$, shows that

$$V_1^{\boldsymbol{\pi}_{t+1}, \boldsymbol{\mathcal{R}}}(s_1) - V_1^{\boldsymbol{\pi}_t, \boldsymbol{\mathcal{R}}}(s_1) = \sum_{h=1}^{H} \sum_{s \in \mathcal{S}} \mu_h^{s_1, \boldsymbol{\pi}_{t+1}, \boldsymbol{\mathcal{T}}}(s) \underbrace{\sum_{a \in \mathcal{A}} \pi_{t+1,h}(a|s) A_h^{\boldsymbol{\pi}_t, \boldsymbol{\mathcal{R}}}(s, a)}_{\geqslant 0} \geqslant 0 \,.$$

(This is the part of the proof where we crucially use that reward functions do not vary over time.) Thus,

$$\max_{\boldsymbol{\pi}} V_1^{\boldsymbol{\pi}, \boldsymbol{\mathcal{R}}}(s_1) - V_1^{\boldsymbol{\pi}_T, \boldsymbol{\mathcal{R}}}(s_1) \leqslant \max_{\boldsymbol{\pi}} V_1^{\boldsymbol{\pi}, \boldsymbol{\mathcal{R}}}(s_1) - \frac{1}{T} \sum_{t=1}^{T} V_1^{\boldsymbol{\pi}_t, \boldsymbol{\mathcal{R}}}(s_1) \leqslant \frac{H(H+1) B_{T,A}}{T} \,,$$

where we applied Theorem 1 for the final bound. $\square$

**Remark 2.** *An inspection of the proof above shows that what actually matters is only that* mean *reward functions $\boldsymbol{r}_t = (r_{t,h})_{h \in [H]}$ be constant over time. Indeed, the value and advantage functions only depend on the $\boldsymbol{\mathcal{R}}_t$ through the $\boldsymbol{r}_t$; this fact is also illustrated in the proof of the performance difference lemma which only requires identical mean reward functions, not the identity of reward functions.*

## 5 Extension 2: Stronger forms of regret

**Contributions of this section.** *We push the logic of the reduction of the control of MDPs to adversarial learning, and leverage stronger forms of regret in adversarial learning. This section thus presents new regret criteria for learning MDPs.*

Definition 1 considers the simplest definition of adversarial regret. However, several stronger notions of regrets were proposed in the literature. The proof of Theorem 1 shows that the vanilla notion of adversarial regret of Definition 1 may be transferred into the vanilla regret (2) in terms of value functions. Actually, this proof may be mimicked to transfer stronger notions of adversarial regret. We illustrate this possibility with two notions of adversarial regrets that replace the comparison to a single global policy with local comparisons (strongly adaptive regret) or by global comparisons to sequences of policies (tracking regret).

### 5.1 Strongly adaptive regret and tracking regret in adversarial learning

We use again the notation for adversarial learning introduced at the beginning of Section 2.1. The first extended notion of regret, called strongly adaptive regret, measures performance simultaneously over each given sub-interval of time with respect to the best component over that sub-interval. It was introduced by Daniely et al. (2015), based on the concept of adaptive regret from Hazan & Seshadhri (2009), itself based on the work by Littlestone & Warmuth (1994).

**Definition 3** (strongly adaptive regret in adversarial learning)**.** *A sequential strategy controls the strongly adaptive regret in the adversarial setting with rewards bounded by $M > 0$ if there exist positive numbers $B_{T,K,\tau}$, where $T \geqslant 1$ and $\tau \in [T]$, such that, against all opponent players sequentially picking reward vectors in $[-M, M]^K$,*

$$\forall T \geqslant 1, \quad \forall \tau \in [T], \qquad \max_{t_0 \in [T-\tau+1]} \left\{ \max_{k \in [K]} \sum_{t=t_0}^{t_0+\tau-1} g_{t,k} - \sum_{t=t_0}^{t_0+\tau-1} \sum_{j \in [K]} w_{t,j} \, g_{t,j} \right\} \leqslant 2M \, B_{T,K,\tau} \,,$$

$$and \quad \sup_{\tau \in [T]} \frac{B_{T,K,\tau}}{T} \to 0 \quad as \ T \to \infty.$$

It follows from Daniely et al. (2015, Theorem 1) that the strongly adaptive regret can be controlled with bounds $B_{T,K,\tau}$ of order $\sqrt{\tau}$ up to logarithmic factors.

A closely related notion is the tracking regret, introduced by Herbster & Warmuth (1998) (see also Cesa-Bianchi & Lugosi, 2006, Chapter 5.2), where the comparison is taken over all time steps but against sequences $k_{1:T} = (k_1, k_2, \ldots, k_T)$ with values in $[K]$, and containing at most $C$ shifts (i.e., $C$ time steps such that $k_t \neq k_{t-1}$). The tracking regret involves

$$\sum_{t=1}^{T} g_{t,k_t} - \sum_{t=1}^{T} \sum_{j \in [K]} w_{t,j} \, g_{t,j} \, .$$

There are strong links between strongly adaptive and tracking regret, see Adamskiy et al. (2016). In particular, we explain, in the context of regret with value functions, how strongly adaptive regret with $B_{T,K,\tau}$ of order $\sqrt{\tau}$ up to logarithmic factors entails tracking regret of order $\sqrt{CT}$; see Corollary 1.

## 5.2 Transfer of strongly adaptive regret bounds

Based on Definition 3, we obtain the following regret bound in terms of value functions and policies.

**Theorem 3.** *In the setting of Section 2 where rewards lie in $[0,1]$, if, for all $h \in [H]$, the sequential strategies $\varphi_h$ control the strongly adaptive regret in the adversarial setting (Definition 3) by $B_{T,A,\tau}$ for $A$–dimensional reward vectors bounded by $H - h + 1$, then the $(\varphi_h)_{h \in [H]}$–$\mathtt{Adv2}$ strategy defined in (6) ensures that*

$$\forall T \geqslant 1, \quad \forall \tau \in [T], \qquad \max_{t_0 \in [T-\tau+1]} \left\{ \max_{\boldsymbol{\pi}} \sum_{t=t_0}^{t_0+\tau-1} \left( V_1^{\boldsymbol{\pi},\mathcal{R}_t}(s_1) - V_1^{\boldsymbol{\pi}_t,\mathcal{R}_t}(s_1) \right) \right\} \leqslant H(H+1) \, B_{T,A,\tau} \, .$$

The proof of Theorem 3 is obtained by a direct adaptation of the proof of Theorem 1, which basically consists of considering sums over sub-intervals only instead of sums over all time periods. Again, since the proof is concise, we provide it here.

*Proof of Theorem 3.* We fix a stationary policy $\boldsymbol{\pi}$ throughout the proof and control some adaptive regret with respect to this $\boldsymbol{\pi}$. By the design (6) of the $\mathtt{Adv2}$ strategy, which operates stage by stage and state by state, we have that for all $h \in [H]$ and $s \in \mathcal{S}$, the following holds, by Definition 3: for all $T \geqslant 1$ and $\tau \in [T]$,

$$\max_{t_0 \in [T-\tau+1]} \left\{ \max_{a \in \mathcal{A}} \sum_{t=t_0}^{t_0+\tau-1} A_h^{\boldsymbol{\pi}_t,\mathcal{R}_t}(s,a) - \sum_{t=t_0}^{t_0+\tau-1} \overbrace{\sum_{a \in \mathcal{A}} \pi_{t,h}(a|s) \, A_h^{\boldsymbol{\pi}_t,\mathcal{R}_t}(s,a)}^{=0} \right\} \leqslant 2(H-h+1) \, B_{T,A,\tau} \, .$$

The same application of the performance difference lemma as in the proof of Theorem 1 entails that for all $T \geqslant 1$, $\tau \in [T]$, and $t_0 \in [T-\tau+1]$,

$$\sum_{t=t_0}^{t_0+\tau-1} \left( V_1^{\boldsymbol{\pi},\mathcal{R}_t}(s_1) - V_1^{\boldsymbol{\pi}_t,\mathcal{R}_t}(s_1) \right) = \sum_{h=1}^{H} \sum_{s \in \mathcal{S}} \mu_h^{s_1,\boldsymbol{\pi},\mathcal{T}}(s) \sum_{a \in \mathcal{A}} \pi_h(a|s) \sum_{t=t_0}^{t_0+\tau-1} A_h^{\boldsymbol{\pi}_t,\mathcal{R}_t}(s,a)$$

$$\leqslant \sum_{h=1}^{H} \sum_{s \in \mathcal{S}} \mu_h^{s_1,\boldsymbol{\pi},\mathcal{T}}(s) \underbrace{\max_{a \in \mathcal{A}} \sum_{t=t_0}^{t_0+\tau-1} A_h^{\boldsymbol{\pi}_t,\mathcal{R}_t}(s,a)}_{\leqslant 2(H-h+1) \, B_{T,A,\tau}} \leqslant H(H+1) \, B_{T,A,\tau} \, .$$

Here again, we crucially used that the weights $\mu_h^{s_1,\boldsymbol{\pi},\mathcal{T}}(s)$ are independent of $t$. The claimed bound follows by taking the maximum over $\boldsymbol{\pi}$ and over $t_0 \in [T-\tau+1]$. $\qquad \square$

### 5.3 Tracking regret bounds

We detail a consequence of the bound of Theorem 3 in terms of tracking regret.

We now consider sequences $\boldsymbol{\pi}^{(1:T)} = \big(\boldsymbol{\pi}^{(1)}, \boldsymbol{\pi}^{(2)}, \ldots, \boldsymbol{\pi}^{(T)}\big)$ of stationary policies as comparison points, instead of a single stationary policy. We define the number of shifts $c\big(\boldsymbol{\pi}^{(1:T)}\big)$ of such a sequence as follows: the smallest integer $c'$ such that there exist $c' - 1$ integers $\tau_2, \ldots, \tau_{c'}$ with values in $[T]$ such that, denoting $\tau_1 = 1$ and $\tau_{c'+1} = T + 1$,

$$\forall i \in \{2, \ldots, c' + 1\}, \qquad \forall t \in \{\tau_{i-1}, \ldots, \tau_i - 1\}, \qquad \boldsymbol{\pi}^{(t)} = \boldsymbol{\pi}^{(\tau_{i-1})}. \tag{10}$$

The tracking regret against sequences $\boldsymbol{\pi}^{(1:T)}$ of stationary policies with at most $C$ shifts is defined as

$$\max_{\substack{\boldsymbol{\pi}^{(1:T)} \text{ such that} \\ c(\boldsymbol{\pi}^{(1:T)}) \leqslant C}} \sum_{t=1}^{T} V_1^{\boldsymbol{\pi}^{(t)}, \boldsymbol{\mathcal{R}}_t}(s_1) - \sum_{t=1}^{T} V_1^{\boldsymbol{\pi}_t, \boldsymbol{\mathcal{R}}_t}(s_1).$$

We fix $1 \leqslant C \leqslant T$ and a sequence $\boldsymbol{\pi}^{(1:T)}$ of stationary policies, with at most $C$ shifts, occurring at episodes $1 = \tau_1 \leqslant \tau_2 \leqslant \ldots \leqslant \tau_C$. (The inequalities are strict if there are exactly $C$ shifts.) We introduce $\tau_{C+1} = T + 1$ and partition time into the $C$ intervals $[\tau_i, \tau_{i+1} - 1]$, for $i \in [C]$. The values successively taken by the sequence $\boldsymbol{\pi}^{(1:T)}$ consist of the $\boldsymbol{\pi}^{(\tau_i)}$, where $i \in [C]$. By applying the bound of Theorem 3 on each of the $C$ intervals $[\tau_i, \tau_{i+1} - 1]$, we obtain the following corollary.

**Corollary 1.** *Under the assumptions of Theorem 3, the $(\varphi_h)_{h \in [H]}$–Adv2 strategy defined in (6) also ensures that* $\quad \forall T \geqslant 1, \quad \forall C \in [T],$

$$\max_{\substack{\boldsymbol{\pi}^{(1:T)} \text{ such that} \\ c(\boldsymbol{\pi}^{(1:T)}) \leqslant C}} \sum_{t=1}^{T} V_1^{\boldsymbol{\pi}^{(t)}, \boldsymbol{\mathcal{R}}_t}(s_1) - \sum_{t=1}^{T} V_1^{\boldsymbol{\pi}_t, \boldsymbol{\mathcal{R}}_t}(s_1) \leqslant H(H+1) \max_{\substack{1 = \tau_1 \leqslant \tau_2 \leqslant \ldots \\ \leqslant \tau_C \leqslant \tau_{C+1} = T+1}} \sum_{i=1}^{C} B_{T, A, \tau_{i+1} - \tau_i}.$$

In particular, if $B_{T,A,\tau} \leqslant \ell(T,K)\sqrt{\tau}$, where $\ell(T,K)$ is logarithmic in $T$ and $K$, which is a standard bound, then by Jensen's inequality for $\sqrt{\cdot}$,

$$\max_{\substack{1 = \tau_1 \leqslant \tau_2 \leqslant \ldots \\ \leqslant \tau_C \leqslant \tau_{C+1} = T+1}} \sum_{i=1}^{C} B_{T, A, \tau_{i+1} - \tau_i} \leqslant \ell(T,K) \max_{\substack{1 = \tau_1 \leqslant \tau_2 \leqslant \ldots \\ \leqslant \tau_C \leqslant \tau_{C+1} = T+1}} \underbrace{\sum_{i=1}^{C} \sqrt{\tau_{i+1} - \tau_i}}_{\leqslant \sqrt{C(\tau_{C+1} - \tau_1)} = \sqrt{CT}} \leqslant \ell(T,K)\sqrt{CT}.$$

## 6 The special case of exponential weights: improved regret bounds

**Contributions of this section.** *The literature (see Section 1.1) essentially focuses on the adversarial learning strategy given by exponential weights with a constant learning rate $\eta$. It turns out that this strategy does not satisfy the requirement of Definition 1 because of a tuning issue: the adversarial regret bound is of the form $\ln N/\eta + \eta MT/2$ (see, e.g., Cesa-Bianchi & Lugosi, 2006, Theorem 2.2) and cannot be simultaneously optimized for all values of $T$. The literature typically assumes that $T$ is known and obtains a $\sqrt{T}$ regret bound for MDPs by taking $\eta$ of order $1/\sqrt{T}$; see, for instance, among many others, Cai et al. (2020) and Shani et al. (2020). A notable exception, in the discounted setting and for a constant reward function, can be extracted from the proof of Agarwal et al. (2021, Section 5.3)—they handle convergence of the last iterate but their proof technique also applies to cumulative regret. We extend their result to the episodic setting and show that it is not essential that the reward functions be constant over time: we provide an upper bound in terms of the numbers of shifts in the sequence of reward functions.*

We study in this section the strategy (6) of Section 3.2 where the adversarial learning strategies are given by the strategy (5) based on a constant exponential potential $\Phi_t \equiv \Phi : x \mapsto \exp(\eta x)$. This strategy takes

the following simple form: for all $t \geqslant 1$,

$$\forall h \in [H], \quad \forall s \in \mathcal{S}, \quad \forall a \in \mathcal{A}, \qquad \pi_{t,h}(a|s) = \frac{\exp\left(\eta \sum_{\tau=1}^{t-1} A_h^{\boldsymbol{\pi}_\tau, \boldsymbol{\mathcal{R}}_\tau}(s,a)\right)}{\sum_{a' \in \mathcal{A}} \exp\left(\eta \sum_{\tau=1}^{t-1} A_h^{\boldsymbol{\pi}_\tau, \boldsymbol{\mathcal{R}}_\tau}(s,a')\right)}, \tag{11}$$

with the understanding that a sum over no term is null, i.e., $\pi_{1,h}(a|s) = 1/A$.

Agarwal et al. (2021, Section 5.3) showed that the strategy above corresponds to the natural policy gradient [NPG] strategy based on a softmax parametrization. They proposed a direct analysis (in the discounted setting) with reward functions constant over time. We adapt and extend this analysis to (obliviously) adversarial sequences of reward functions. We also claim a more transparent proof scheme, consisting of a suitable adversarial bound (finer than the uniform bounds considered in Definition 1, which in this case would be linear in $T$, as recalled in the introduction of this section) applied to policy learning along the lines of the proof of Theorem 1.

Our result is stated in terms of the number $R$ of regimes shifts in the sequence $\boldsymbol{\mathcal{R}}_1, \ldots, \boldsymbol{\mathcal{R}}_T$ of payoff functions. More formally, $R$ is the smallest integer such that there exist $R-1$ integers $\tau_2, \ldots, \tau_R$ with values in $[T]$ such that, denoting $\tau_1 = 1$ and $\tau_{R+1} = T + 1$,

$$\forall k \in \{2, \ldots, R+1\}, \qquad \forall t \in \{\tau_{k-1}, \ldots, \tau_k - 1\}, \qquad \boldsymbol{\mathcal{R}}_t = \boldsymbol{\mathcal{R}}_{\tau_{k-1}}. \tag{12}$$

(The case $R = 1$ corresponds to a single regime, i.e., the reward functions $\boldsymbol{\mathcal{R}}_t$ are independent of time.)

The proof of Theorem 4 below may be found in Appendix A. It is more complex than the proof by Agarwal et al. (2021, Section 5.3), which could use a simple argument specific to the discounted setting, with discount factor $\gamma$: that distributions over states induced by a starting state $s_0$, a policy, and a transition function, put a probability mass at least $1 - \gamma$ on $s_0$, no matter the policy and the transition function. See Remark 6 for more details.

**Theorem 4.** *In the setting of Section 2 where rewards lie in $[0,1]$, the policy learning strategy* (11) *controls the regret as*

$$\max_{\boldsymbol{\pi}} \sum_{t=1}^{T} \left( V_1^{\boldsymbol{\pi}, \boldsymbol{\mathcal{R}}_t}(s_1) - V_1^{\boldsymbol{\pi}_t, \boldsymbol{\mathcal{R}}_t}(s_1) \right) \leqslant \frac{H \ln A}{\eta} + R \frac{H(H+1)}{2},$$

*where $R$ is the number of regime shifts in the sequence $\boldsymbol{\mathcal{R}}_1, \ldots, \boldsymbol{\mathcal{R}}_T$ of payoff functions.*

The bound of Theorem 4 has a smaller order of magnitude than the one of Theorem 1, which is typically of order $\sqrt{T}$, as soon as the number of regime shifts satisfies $R \ll \sqrt{T}$. (In general, up to $T - 1$ regime shifts may occur.) In particular, the regret upper bound of Theorem 4 is smaller than a constant when the reward functions do not vary over time. Of course, as already mentioned at the beginning of Section 3.2, this observation is somewhat secondary in the absence of an oracle for value functions, when value functions have to be estimated and when these estimation errors are the main contributors to the regret bounds; see Section 8.

By Lemma 2 and (the proof of) Theorem 2, we have the following corollary to Theorem 4, in case of a constant sequence of payoff functions. It corresponds to the bound of Agarwal et al. (2021, Section 5.3) with $H$ playing the role of $1/(1 - \gamma)$ therein.

**Corollary 2.** *In the setting of Section 2 where rewards lie in $[0,1]$, if the reward functions do not vary over time and are all equal to some $\boldsymbol{\mathcal{R}}$, then the last iterate of the policy learning strategy* (11) *satisfies*

$$\max_{\boldsymbol{\pi}} V_1^{\boldsymbol{\pi}, \boldsymbol{\mathcal{R}}}(s_1) - V_1^{\boldsymbol{\pi}_T, \boldsymbol{\mathcal{R}}}(s_1) \leqslant \frac{H \ln A}{\eta T} + \frac{H(H+1)}{2T}.$$

As in Agarwal et al. (2021, Section 5.3), the bounds obtained in Theorem 4 and Corollary 2 suggest choosing $\eta$ as large as possible. While this is the choice recommended by theory, practical performance may be affected: in Section 8, we recommend to conduct empirical evaluations to investigate this issue.

# 7 Extension 3: Aggregation (orchestration) of expert policies

**Contributions of this section.** *Adversarial learning is sometimes called prediction with experts (see Cesa-Bianchi & Lugosi, 2006). We further pursue the idea of the reduction of the control of MDPs to adversarial learning and now rather aggregate expert policies. The aim is to mimic the performance of the overall best convex combination of expert policies (which is, in particular, better than the performance of the best policy taken in isolation), which corresponds to an aggregation (or orchestration) of expert policies. This setting was also termed learning from multiple oracles (which may be understood as a specific paradigm in the vast imitation-learning literature) by Cheng et al. (2020) and Liu et al. (2023). We obtain stronger forms of performance guarantees than in the latter references, see Remark 3. We do so via some reduction to the standard tabular case for a lifted MDP.*

We return to the considerations of Section 2.1 and consider a finite number $K$ of stationary policies. We denote by $\Pi = \{\boldsymbol{\pi}_1, \ldots, \boldsymbol{\pi}_K\}$ the set of these policies and refer to them as expert policies. Furthermore, for a given stage $h \in [H]$, we denote by $\Pi_h = \{\pi_{1,h}, \ldots, \pi_{K,h}\}$ the set of the corresponding policies.

We combine expert policies over time through state-stage-dependent weights $\boldsymbol{p}_t = (p_{t,h})_{h\in[H]} \in \mathcal{P}\big([K]\big)^{[H]\times\mathcal{S}}$, where $p_{t,h}(\,\cdot\,|s) \in \mathcal{P}\big([K]\big)$ may be interpreted either as a probability distribution over the policies in $\Pi_h$ or as providing convex weights for the aggregation of the policies in $\Pi_h$. More precisely, for each episode $t \geqslant 1$, we denote by $\boldsymbol{p}_t\Pi = (p_{t,h}\Pi_h)_{h\in[H]}$ the stationary policy such that, for all stages $h \in [H]$,

$$p_{t,h}\Pi_h : s \in \mathcal{S} \longmapsto p_{t,h}\Pi_h(\,\cdot\,|s) = \sum_{k\in[K]} p_{t,h}(k|s)\,\pi_{k,h}(\,\cdot\,|s) \in \mathcal{P}(\mathcal{A})\,. \tag{13}$$

Picking an action $a'$ according to $p_{t,h}\Pi_h(\,\cdot\,|s)$ amounts to performing a two-stage randomization: first, drawing a policy index $k' \sim p_{t,h}(\,\cdot\,|s)$, then drawing $a' \sim \pi_{k',h}(\,\cdot\,|s)$. This remark is important in the cases where it is difficult or computationally complex to explicitly write the $\pi_{k,h}(\,\cdot\,|s)$, but where it is easy to simulate them.

As indicated above, the set of all possible state-stage-dependent weights $\boldsymbol{q}$ corresponds to $\mathcal{P}\big([K]\big)^{[H]\times\mathcal{S}}$. We consider the class $\mathcal{C}(\Pi)$ of all possible stationary policies defined according to (13):

$$\mathcal{C}(\Pi) = \Big\{\boldsymbol{q}\Pi, \ \boldsymbol{q} \in \mathcal{P}\big([K]\big)^{[H]\times\mathcal{S}}\Big\}\,,$$

and aim to learn a good policy in this class. To do so, the learning strategies pick weights $\boldsymbol{p}_t \in \mathcal{P}\big([K]\big)^{[H]\times\mathcal{S}}$ over time and output $\boldsymbol{\pi}_t = \boldsymbol{p}_t\Pi$. We will minimize the corresponding regret criterion:

$$\forall T \geqslant 1, \qquad R_T^\Pi = \max_{\boldsymbol{q}} \sum_{t=1}^{T} \Big(V_1^{\boldsymbol{q}\Pi,\boldsymbol{\mathcal{R}}_t}(s_1) - V_1^{\boldsymbol{p}_t\Pi,\boldsymbol{\mathcal{R}}_t}(s_1)\Big)\,.$$

**Remark 3.** *To the best of our understanding, the recent contributions by Cheng et al. (2020) and Liu et al. (2023) mentioned above consider a more restrictive setting with a constant reward function and, in addition, target a weaker notion of regret, corresponding to*

$$\max_{k\in[K]} V_1^{\boldsymbol{\delta}_k\Pi,\boldsymbol{\mathcal{R}}}(s_1) - \max_{t\in[T]} V_1^{\boldsymbol{p}_t\Pi,\boldsymbol{\mathcal{R}}}(s_1)\,,$$

*where each $\boldsymbol{\delta}_k$ is a collection of state-stage-dependent weights that are all given by Dirac masses on expert $k$; i.e., $V_1^{\boldsymbol{\delta}_k\Pi,\boldsymbol{\mathcal{R}}} = V_1^{\boldsymbol{\pi}_k,\boldsymbol{\mathcal{R}}}$.*

Actually, the total regret $R_T$ defined in Section 2.1 may be decomposed into some approximation error, i.e., how good the policies in $\mathcal{C}(\Pi)$ are in terms of values, plus the regret with respect to $\mathcal{C}(\Pi)$:

$$R_T = \max_{\boldsymbol{\pi}} \sum_{t=1}^{T} \Big(V_1^{\boldsymbol{\pi},\boldsymbol{\mathcal{R}}_t}(s_1) - V_1^{\boldsymbol{\pi}_t,\boldsymbol{\mathcal{R}}_t}(s_1)\Big) = \underbrace{\max_{\boldsymbol{\pi}} \sum_{t=1}^{T} V_1^{\boldsymbol{\pi},\boldsymbol{\mathcal{R}}_t}(s_1) - \max_{\boldsymbol{q}} \sum_{t=1}^{T} V_1^{\boldsymbol{q}\Pi,\boldsymbol{\mathcal{R}}_t}(s_1)}_{\text{approximation error}} + R_T^\Pi\,.$$

In this section, we aim to control $R_T^\Pi$ only and will assume that the approximation error is small due to a proper choice of $\Pi$. This situation is expected to arise frequently, as explained in the following remark.

**Remark 4.** *Denote by $\boldsymbol{\pi}^\star$ a stationary policy achieving the maximum in the definition of $R_T$. Given that expert policies are combined through state-stage-dependent weights, the approximation error defined above is null as soon as*

$$\forall h \in [H], \ \forall s \in \mathcal{S}, \quad \exists q_h(\cdot\,|s) \in \mathcal{P}\big([K]\big) \quad s.t. \quad \pi_h^\star(\cdot\,|s) = \sum_{k \in [K]} q_h(k|s)\,\pi_{k,h}(\cdot\,|s)\,.$$

*In particular, it suffices that there exists $j_{h,s}^\star \in [K]$ such that $\pi_h^\star(\cdot\,|s) = \pi_{j_{h,s}^\star,h}(\cdot\,|s)$. Put differently, it suffices that at each stage $h \in [H]$ and for each state $s \in \mathcal{S}$, one of the expert policies (but not necessarily always the same) coincides with an optimal policy. This observation motivates the use of expert policies in the cases where finitely many easy-to-identify distributions are candidates to be optimal distributions for each given stage-state pair $(h, s)$.*

**Summary.** We provide in Box B a summary of the settings and aims considered, here in Section 7 and earlier in Section 2.1 (the left-hand side of Box B corresponds to Box A of Section 2.1).

### 7.1 Equivalence between direct tabular learning and aggregation of expert policies

We now explain why any learning scheme minimizing the standard regret $R_T$ induces a learning scheme minimizing the regret $R_T^\Pi$ with respect to a finite set $\Pi$ of expert policies, and vice versa. In a nutshell, the equivalence stems from considering the indexes $k \in [K]$ of expert policies as meta-actions, i.e., actions in a sequence of lifted MDPs.

As a consequence, for the sake of clarity and completeness, we will re-state the counterpart of our main result, Theorem 1, in the setting of policy orchestration: see Section 7.2. For now, we prove the claimed equivalence.

**Direct tabular learning as aggregation of expert policies.** We set $K = A$ and take as expert policies the Dirac masses on the arms; more precisely, for each $a \in \mathcal{A}$, and for all $h \in [H]$ and $s \in \mathcal{S}$, we set $\pi_{a,h}(\cdot\,|s) = \delta_a$, the Dirac mass at $a$. This defines the expert policy $\boldsymbol{\Delta}_a$. We consider

$$\Delta = \{\boldsymbol{\Delta}_a : a \in \mathcal{A}\} \qquad \text{and} \qquad \mathcal{C}(\Delta) = \big\{\boldsymbol{p}\Delta, \ \boldsymbol{p} \in \mathcal{P}(\mathcal{A})^{[H]\times\mathcal{S}}\big\}\,;$$

$\mathcal{C}(\Delta)$ is the set of all stationary policies, stated in their direct tabular form.

**From direct tabular learning to aggregation of expert policies.** Conversely, we note that aggregation of expert policies in $\Pi$ amounts to performing direct tabular learning in the following sequence of (lifted) MDPs: the action space is $\overline{\mathcal{A}} = [K]$, the state space is $\overline{\mathcal{S}} = \mathcal{S}$, the transition kernels $\overline{\mathcal{T}}$ and the reward functions $\overline{\mathcal{R}}_t$ are defined, for all $t \geqslant 1$ and $h \in [H]$, by

$$\overline{\mathcal{T}}_h : (s,k) \in \mathcal{S}\times[K] \longmapsto \sum_{a\in\mathcal{A}} \pi_{k,h}(a|s)\,\mathcal{T}_h(\cdot\,|s,a)$$

$$\text{and} \qquad \overline{\mathcal{R}}_{t,h} : (s,k) \in \mathcal{S}\times[K] \longmapsto \sum_{a\in\mathcal{A}} \pi_{k,h}(a|s)\,\mathcal{R}_{t,h}(s,a)\,.$$

Direct tabular learning on the sequence of lifted MDPs defined above provides policies $\overline{\boldsymbol{\pi}}_t$ which correspond to the convex weights $\boldsymbol{p}_t$ discussed above: for all $t \geqslant 1$, $h \in [H]$, and $s \in \mathcal{S}$, we use $p_{t,h}(\cdot\,|s) = \overline{\pi}_{t,h}(\cdot\,|s)$ to aggregate expert policies in the original MDP. Denoting by $\overline{R}_T$ the regret suffered with direct tabular learning in the lifted MDP, we have: $R_T^\Pi = \overline{R}_T$.

**Remark 5.** *In the final part of the proof of Theorem 1, we critically used that the transition kernels $\mathcal{T}$ do not depend on time. The expression above for $\overline{\mathcal{T}}$ is indeed independent on time, which would not be the case if the expert policies were evolving over time. This explains why we restricted our attention to stationary expert policies.*

---

**BOX B: POLICY OPTIMIZATION, POSSIBLY BASED ON EXPERT POLICIES**

| Direct tabular learning (Section 2.1) | Aggregation of expert policies (Section 7) |

**MDP parameters:** state space $\mathcal{S}$, action space $\mathcal{A}$, initial state $s_1 \in \mathcal{S}$, transition kernels $\mathcal{T}$

| *(No additional parameters)* | Set $\Pi$ of $K$ expert policies |

The environment picks a sequence $(\mathcal{R}_t)_{t \geqslant 1}$ of reward functions

**For episodes** $t = 1, 2, \ldots$:

1. The initial state is set to $s_{t,1} = s_1$

2. **For stages** $h = 1, \ldots, H$:

| (a) The learner picks a policy $\pi_{t,h} : \mathcal{S} \to \mathcal{P}(\mathcal{A})$ 
 (b) and draws an action $a_{t,h} \sim \pi_{t,h}(\,\cdot\,|s_{t,h})$ | (a) The learner picks weights $p_{t,h} \in \mathcal{P}([K])^{\mathcal{S}}$, 
 (b) draws $k_{t,h} \sim p_{t,h}(\,\cdot\,|s_{t,h})$, the index of the expert policy, 
 (c) and draws an action $a_{t,h} \sim \pi_{k_{t,h},h}(\,\cdot\,|s_t)$ according to expert policy $k_{t,h}$ |

4. The learner receives and observes a reward drawn independently from $\mathcal{R}_{t,h}(s_{t,h}, a_{t,h})$, with conditional expectation $r_{t,h}(s_{t,h}, a_{t,h})$

5. If $h \leqslant H - 1$, the next state $s_{t,h+1} \sim \mathcal{T}_h(\,\cdot\,|s_{t,h}, a_{t,h})$ is drawn

**Goal:** Minimize the regret

| $$R_T = \max_{\boldsymbol{\pi}} \sum_{t=1}^{T} \left( V_1^{\boldsymbol{\pi}, \mathcal{R}_t}(s_1) - V_1^{\boldsymbol{\pi}_t, \mathcal{R}_t}(s_1) \right)$$ | $$R_T^{\Pi} = \max_{\boldsymbol{q}} \sum_{t=1}^{T} \left( V_1^{\boldsymbol{q}\Pi, \mathcal{R}_t}(s_1) - V_1^{\boldsymbol{p}_t\Pi, \mathcal{R}_t}(s_1) \right)$$ |

---

### 7.2 Adversarial learning on advantage functions for aggregation of expert policies

The counterpart for aggregation of expert policies of the strategy defined in Section 3.2 is defined as follows, given the equivalence stated above.

For each stage $h \in [H]$, we fix a sequential strategy $\varphi_h = (\varphi_{t,h})_{t \geqslant 1}$ in the adversarial setting, relying on reward vectors bounded by $M_h = H - h + 1$ and of dimension $K$.

We run these strategies on the advantage functions of the lifted MDPs described above: for all $t \geqslant 1$, $h \in [H]$, and $s \in \mathcal{S}$,

$$\overline{A}_h^{\boldsymbol{p}_t, \overline{\mathcal{R}}_t}(s, \cdot) = \left( \overline{A}_h^{\boldsymbol{p}_t, \overline{\mathcal{R}}_t}(s, k) \right)_{k \in [K]}, \qquad \text{where} \qquad \overline{A}_h^{\boldsymbol{p}_t, \overline{\mathcal{R}}_t}(s, k) = \sum_{a \in \mathcal{A}} \pi_{k,h}(a|s) \, A_h^{\boldsymbol{p}_t\Pi, \mathcal{R}_t}(s, a) \,. \qquad (14)$$

More precisely, we run the strategies $(\varphi_h)_{h \in [H]}$ in the following stage-by-stage and state-by-state manner: for all $t \geqslant 1$,

$$p_{t,h}(\,\cdot\,|s) = \varphi_{t,h}\left( \left( \overline{A}_h^{\boldsymbol{p}_\tau, \overline{\mathcal{R}}_\tau}(s, \cdot) \right)_{\tau \leqslant t-1} \right) \,. \qquad (15)$$

We refer to this strategy as $(\varphi_h)_{h \in [H]}$–$\texttt{Adv2-Aggr}$, for $(\varphi_h)_{h \in [H]}$–adversarial learning on advantage functions for aggregation of expert policies.

Theorem 1 immediately entails the following performance guarantee, given the equivalence proved in Section 7.1.

**Corollary 3.** *In the setting of Section 7 where rewards lie in $[0,1]$, if, for all $h \in [H]$, the sequential strategies $\varphi_h$ control the regret in the adversarial setting (Definition 1) by $B_{T,K}$ for $K$–dimensional reward vectors bounded by $H - h + 1$, then the $(\varphi_h)_{h \in [H]}$–$\texttt{Adv2-Aggr}$ strategy defined in* (15) *over the set $\Pi$ of $K$ expert policies controls the regret with respect to $\mathcal{C}(\Pi)$ as:*

$$\forall T \geqslant 1, \qquad R_T^\Pi = \max_{\boldsymbol{q}} \sum_{t=1}^{T} \left( V_1^{\boldsymbol{q}\Pi, \boldsymbol{\mathcal{R}}_t}(s_1) - V_1^{\boldsymbol{p}_t \Pi, \boldsymbol{\mathcal{R}}_t}(s_1) \right) \leqslant H(H+1)\, B_{T,K}\,.$$

## 8 Empirical impacts as future research directions

**Contributions of this section.** *We review in greater detail how the literature resorted or should resort to adversarial learning strategies in practice: value functions are typically not observed and must be estimated. These considerations call for future empirical research.*

This final section relaxes the assumption of an oracle providing value functions, as stated at the beginning of Section 3.2, and first recalls how the literature (see, e.g., Abbasi-Yadkori et al., 2019, Shani et al., 2020, Cai et al., 2020, He et al., 2022, Zhao et al., 2023) typically performs the policy-improvement step for learning adversarial MDPs: by resorting to the exponential-weight strategy of Section 6 based on estimated $Q$–values. We provide a concrete example in Figure 1.

More precisely, the most popular approach is to build optimistic estimates $\widehat{Q}_h^t$ of the true $Q$–value functions $Q_h^{\boldsymbol{\pi}_t, \boldsymbol{\mathcal{R}}_t}$, i.e., estimates that upper bound the true values with high probability. The main issue in doing so is that the transition kernels $\boldsymbol{\mathcal{T}}$ are unknown; whether the reward functions $\boldsymbol{\mathcal{R}}_t$ are fully revealed (full-information feedback) or not (bandit feedback, where only actual rewards are observed) at the end of an episode may be handled (see, among others, Shani et al., 2020). Also, these optimistic estimates $\widehat{Q}_h^t$ may or may not rely on structural assumptions (e.g., Cai et al., 2020 and He et al., 2022 assume some linear representation of the transition kernels) and are specific to each article mentioned.

However, what is common to these articles, is the way the policy-improvement step is performed based on these estimated $Q$–values; this way is illustrated in Figure 1. Abbasi-Yadkori et al. (2019) even states that "the choice of the Boltzmann policy is not arbitrary" in this step (where "Boltzmann policy" is a synonym for the exponential-weight strategy).

*The point of the present article is exactly to question this common practice and show that other choices are possible* (as detailed in Section 8.1), *with no impact on the theoretical guarantees proved in these articles* (see Section 8.2). *These observations call for future empirical research* (see Section 8.3).

### 8.1 Other choices for the policy-improvement step typically considered in the literature

As stated above and illustrated in Figure 1, the literature on adversarial MDPs (see, e.g., Abbasi-Yadkori et al., 2019, Shani et al., 2020, Cai et al., 2020, He et al., 2022, Zhao et al., 2023) typically considers policy-improvement steps of the form, for some $\eta > 0$,

$$\forall t \geqslant 1, \quad \forall h \in [H], \quad \forall s \in \mathcal{S}, \quad \forall a \in \mathcal{A},$$

$$\pi_{t+1,h}(a|s) = \frac{\pi_{t,h}(a|s)\, \exp\!\big(\eta\,\hat{Q}_h^t(s,a)\big)}{\sum\limits_{a' \in \mathcal{A}} \pi_{t,h}(a'|s)\, \exp\!\big(\eta\,\hat{Q}_h^t(s,a')\big)} = \frac{\exp\!\left(\eta \sum\limits_{\tau=1}^{t} \hat{Q}_h^\tau(s,a)\right)}{\sum\limits_{a' \in \mathcal{A}} \exp\!\left(\eta \sum\limits_{\tau=1}^{t} \hat{Q}_h^\tau(s,a')\right)}\,. \tag{16}$$



|  | |
|---|---|
| *Strategy by Shani et al. (2020)* | *Alternative formulations* |

**Algorithm 2** Optimistic POMD for Stochastic MDPs

**Require:** $t_K$, $\pi_1$ is the uniform policy.
  **for** $k = 1, ..., K$ **do**
    Rollout a trajectory by acting $\pi_k$
    # Policy Evaluation
    $\forall s \in \mathcal{S},\ V_{H+1}^k(s) = 0$
    **for** $\forall h = H, .., 1$ **do**
      **for** $\forall s, a \in \mathcal{S} \times \mathcal{A}$ **do**
        $\hat{c}_h^{k-1}(s,a) = \bar{c}_h^{k-1}(s,a) - b_h^{k-1}(s,a)$, Eq. (6.1)
        $Q_h^k(s,a) = \hat{c}_h^{k-1}(s,a) + \bar{p}_h^{k-1}(\cdot|s,a)V_{h+1}^k$
        $Q_h^k(s,a) = \max\{Q_h^k(s,a), 0\}$
      **end for**
      **for** $\forall s \in \mathcal{S}$ **do**
        $V_h^k(s) = \langle Q_h^k(s,\cdot), \pi_h^k(\cdot \mid s) \rangle$
      **end for**
    **end for**
    # Policy Improvement
    **for** $\forall h, s, a \in [H] \times \mathcal{S} \times \mathcal{A}$ **do**
      $\pi_h^{k+1}(a|s) = \dfrac{\pi_h^k(a|s)\exp\left(-t_K Q_h^k(s,a)\right)}{\sum_{a'} \pi_h^k(a'|s)\exp\left(-t_K Q_h^k(s,a')\right)}$
    **end for**
    Update counters and empirical model, $n_k, \bar{c}^k, \bar{p}^k$
  **end for**

Right column:

**for** all $(h,s) \in [H] \times \mathcal{S}$ **do**

$$\pi_h^{k+1}(\cdot|s) = \varphi_{k+1,h}\left(\left(-Q_h^\tau(s,\cdot)\right)_{\tau \leqslant k}\right)$$

or, based on estimated advantage functions defined as in Assumption 1:

$$\pi_h^{k+1}(\cdot|s) = \varphi_{k+1,h}\left(\left(-A_h^\tau(s,\cdot)\right)_{\tau \leqslant k}\right)$$

**end for**

$\longrightarrow$



Figure 1: The strategy considered and studied by Shani et al. (2020), as stated therein (*left part*): our results focus on considering alternative formulations of the policy-improvement step, based on other adversarial-learning strategies than exponential weights, and possibly based on estimated advantage functions rather than estimated $Q$–values (*right part*). Shani et al. (2020) considers costs instead of rewards, hence the negative signs appearing when feeding adversarial learning strategies $\varphi$ designed for rewards.

We propose alternative formulations, based on advantage functions obtained from the $Q$–values as described by Assumption 1. This assumption is satisfied as soon as value functions are also obtained by the same convex combinations, which is the case in all references mentioned above, but would not be the case for other approaches, for instance, for $Q$–learning or similar methods that would define $\widehat{V}_h^t(s)$ as $\max_{a' \in \mathcal{A}} \widehat{Q}_h^t(s,a')$.

Assumption 1 also imposes boundedness of the estimated value functions, as this is key for the theoretical analysis of performance (and is coherent with the fact that the true value functions are also bounded, by known bounds).

**Assumption 1.** *The estimates $\widehat{A}_h^t$ are defined based on estimates $\widehat{Q}_h^t$ of $Q$–value functions, using the policy $\boldsymbol{\pi}_t$ selected: for all $s \in \mathcal{S}$ and $a \in \mathcal{A}$,*

$$\widehat{V}_h^t(s) \overset{\text{def}}{=} \sum_{a' \in \mathcal{A}} \pi_{t,h}(a'|s)\,\widehat{Q}_h^t(s,a') \qquad and \qquad \widehat{A}_h^t(s,a) \overset{\text{def}}{=} \widehat{Q}_h^t(s,a) - \widehat{V}_h^t(s).$$

*In addition, the estimates $\widehat{Q}_h^t$ and $\widehat{A}_h^t$ are bounded, i.e., $0 \leqslant \widehat{Q}_h^t \leqslant M_H$ and $\left|\widehat{A}_h^t\right| \leqslant M_H$ for some quantity $M_H$ (typically depending on $H$).*

Formally, we propose to replace updates of the form (16) with the updates based on the strategy Adv2 as stated in (6), or its variant based on $Q$–values stated in (9):

$$\forall t \geqslant 1, \quad \forall h \in [H], \quad \forall s \in \mathcal{S}, \qquad \pi_{t,h}(\cdot|s) = \varphi_{t,h}\left(\left(\widehat{A}_h^\tau(s,\cdot)\right)_{\tau \leqslant t-1}\right)$$

$$\text{or} \qquad \forall t \geqslant 1, \quad \forall h \in [H], \quad \forall s \in \mathcal{S}, \qquad \pi_{t,h}(\cdot|s) = \varphi_{t,h}\left(\left(\widehat{Q}_h^\tau(s,\cdot)\right)_{\tau \leqslant t-1}\right).$$

Empirically, updates based on estimated advantage functions should perform better.

## 8.2 Preserved theoretical guarantees with these alternative choices

When the sequential strategies $\varphi_h$ control the regret in the adversarial setting (Definition 1) by $B_{T,A}$ for $A$–dimensional reward vectors, the same argument as in (7), together with Assumption 1 for the equality to 0, shows that the strategies defined above satisfy: for all $h \in [H]$ and $s \in \mathcal{S}$,

$$\max_{a \in \mathcal{A}} \sum_{t=1}^{T} \widehat{A}_h^t(s,a) - \sum_{t=1}^{T} \overbrace{\sum_{a \in \mathcal{A}} \pi_{t,h}(a|s) \, \widehat{A}_h^t(s,a)}^{=0} \leqslant 2M_H \, B_{T,A} \,,$$

thus, for all $h \in [H]$ and $s \in \mathcal{S}$,

$$\widehat{R}_T \stackrel{\text{def}}{=} \max_{\boldsymbol{\pi}} \sum_{h=1}^{H} \sum_{s \in \mathcal{S}} \mu_h^{s_1,\boldsymbol{\pi},\boldsymbol{\mathcal{T}}}(s) \sum_{a \in \mathcal{A}} \pi_h(a|s) \sum_{t=1}^{T} \widehat{A}_h^t(s,a) \leqslant 2M_H \, B_{T,A} \,. \tag{17}$$

Specific arguments (see detail below) then relate the quantity above to the target quantity, stated as in (8):

$$R_T = \max_{\boldsymbol{\pi}} \sum_{t=1}^{T} \left( V_1^{\boldsymbol{\pi},\boldsymbol{\mathcal{R}}_t}(s_1) - V_1^{\boldsymbol{\pi}_t,\boldsymbol{\mathcal{R}}_t}(s_1) \right) = \max_{\boldsymbol{\pi}} \sum_{h=1}^{H} \sum_{s \in \mathcal{S}} \mu_h^{s_1,\boldsymbol{\pi},\boldsymbol{\mathcal{T}}}(s) \sum_{a \in \mathcal{A}} \pi_h(a|s) \sum_{t=1}^{T} A_h^{\boldsymbol{\pi}_t,\boldsymbol{\mathcal{R}}_t}(s,a) \,. \tag{18}$$

**Typical examples.** For instance, in the theoretical analysis by Shani et al. (2020, Section 6), the total regret $R_T$ is decomposed as a sum of three terms, where term (ii) therein is exactly (17) but terms (i) and (iii) are bounded in some specific way.

The same may be mentioned for the other references, where the total regret $R_T$ is also decomposed into three terms, with $\widehat{R}_T$ being one of the three terms: in Cai et al. (2020), term (i); in He et al. (2022), term $I_1$; in Zhao et al. (2023), the "OMD regret term".

Again, the adversarial-learning strategy $(\varphi_h)_{h \in [H]}$ considered in all these references is the exponential potential with a constant learning rate (see Section 6), possibly seen as an instance of online mirror descent, and bounds of typical order $\sqrt{T \ln A}$ are achieved on the term corresponding to $\widehat{R}_T$. Thus, considering other adversarial-learning strategies, in the forms described in Section 8.1, would not hurt the final regret bounds achieved on $R_T$, as the errors stemming from the control of $\widehat{R}_T$ are not the main contributors to the final regret bounds achieved in these articles.

**An exception.** In general, $\widehat{R}_T$ is not equal to a sum of differences of value functions. An exception is to be found in Tiapkin et al. (2025): they obtain the estimates $\widehat{Q}_h^t$ as the *exact* $Q$–value functions (obtained by dynamic programming) corresponding to the policies $\boldsymbol{\pi}_t$, to some reward functions $\boldsymbol{\mathcal{R}}_t'$ (based on the actual reward function $\boldsymbol{\mathcal{R}}_t$ revealed at the end of the episode plus some bonus function), and to some estimated transition kernels $\widehat{\boldsymbol{\mathcal{T}}}_t$ (that are constant over subintervals of episodes and are only updated from time to time). Tiapkin et al. (2025) decompose the total regret in four terms, where term (B) corresponds to $\widehat{R}_T$.

Note that Tiapkin et al. (2025) refers to the present work and is therefore able to present the analysis in a more modular way than many of the references mentioned above (some of them re-deriving regret guarantees in terms of $Q$–value functions by mimicking proofs of adversarial regret bounds for exponential weights, as reviewed in Section 1.1).

## 8.3 Future research: evaluation of the empirical impacts of these alternative choices

We explained in Section 8.2 that the modifications proposed in Section 8.1 for the policy-improvement step preserve the theoretical guarantees (essentially because this policy-improvement step is not at all the main blocking point in the analysis).

Yet, a different formulation of this policy-improvement step may dramatically affect the practical performance obtained by these strategies. Many adversarial-learning strategies exist, and exponential-weight strategies are often not the best-performing ones. The `R` package `Opera` by Gaillard et al. (2021) implements several adversarial-learning strategies, some of which (for instance, `ML-Prod` and `ML-Poly`, introduced by Gaillard et al., 2014) often achieving superior empirical performance compared to exponential weights (see the empirical studies in Gaillard, 2015).

Therefore, in our opinion, an interesting avenue of empirical research would be the following. Consider the experimental designs provided by, or create experimental designs corresponding to, the settings of Abbasi-Yadkori et al. (2019), Shani et al. (2020), Cai et al. (2020), He et al. (2022), Zhao et al. (2023). Compare the performance of the strategies introduced therein (with their original policy-improvement steps, based on exponential weights) to alternative strategies differing only in their policy-improvement steps (based on the adversarial-learning strategies implemented in the `Opera` package, used either with estimated $Q$–values or with estimated advantage functions). Also, for exponential weights, as mentioned after Corollary 2, it would be interesting to see the effect of the learning rate $\eta$ on the empirical performance. We leave these studies for future research.

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

**Appendix.** The appendix provides proofs omitted from the main body of the article.

# A Proof of Theorem 4 (analysis of NPG with softmax parametrization)

For the convenience of the reader, we restate the result to be proved.

**Theorem 4.** *In the setting of Section 2 where rewards lie in* $[0, 1]$*, the policy learning strategy* (11) *controls the regret as*

$$\max_{\boldsymbol{\pi}} \sum_{t=1}^{T} \left( V_1^{\boldsymbol{\pi}, \boldsymbol{\mathcal{R}}_t}(s_1) - V_1^{\boldsymbol{\pi}_t, \boldsymbol{\mathcal{R}}_t}(s_1) \right) \leqslant \frac{H \ln A}{\eta} + R \frac{H(H+1)}{2} \,,$$

*where* $R$ *is the number of regime shifts in the sequence* $\boldsymbol{\mathcal{R}}_1, \ldots, \boldsymbol{\mathcal{R}}_T$ *of payoff functions.*

As indicated in Section 6, the proof below is based on the analysis of the natural policy gradient [NPG] with softmax parametrization proposed by Agarwal et al. (2021, Section 5.3) in the discounted setting with reward functions constant over time. See Remark 6 for an explanation of why the proof in the discounted setting is significantly simpler than the proof in the episodic setting.

We extend the proof of Agarwal et al. (2021, Section 5.3) to the episodic setting and to (obliviously) adversarial sequences of reward functions. We also claim a more transparent proof scheme, consisting of an ad hoc adversarial bound (Lemma 3) which is then applied to policy learning along the lines of the proof of Theorem 1.

More precisely, the first piece of the proof of Theorem 4 is to replace the uniform regret bounds considered in Definition 1 with some ad hoc, data-based, bound (of the same flavor as the bounds by de Rooij et al., 2014, Section 2 in terms of so-called mixability gaps). Indeed, the uniform regret bound that could be proved (see, e.g., Cesa-Bianchi & Lugosi, 2006, Theorem 2.2) for the adversarial strategy of Lemma 3 is $B_{T,K} = \ln K / \eta + \eta T / 8$, which is not sublinear.

**Lemma 3.** *The strategy* (5) *based on a constant exponential potential* $\Phi_t \equiv \Phi : x \mapsto \exp(\eta x)$*, i.e., picking weights*

$$\forall t \geqslant 1, \qquad w_{t,k} = \frac{v_{t,k}}{\displaystyle\sum_{j \in [K]} v_{t,j}} \,, \qquad where \qquad v_{t,k} = \exp\left( \eta \sum_{\tau=1}^{t-1} g_{\tau,k} \right),$$

with the convention that $v_{1,k} = 1$ and $w_{1,k} = 1/K$, satisfies the following bound: against all opponents sequentially picking reward vectors in $\mathbb{R}^K$,

$$\forall T \geqslant 1, \qquad \max_{k \in [K]} \sum_{t=1}^T g_{t,k} \leqslant \frac{\ln K}{\eta} + \sum_{t=1}^T \sum_{j \in [K]} w_{t+1,j}\, g_{t,j}\,.$$

This lemma is proved at the end of this section and we now apply it to prove Theorem 4.

*Proof of Theorem 4.* We adapt the proof of Theorem 1 by replacing (7) by the ad hoc bound stemming from Lemma 3; we obtain:

$$\forall h \in [H], \quad \forall s \in \mathcal{S}, \qquad \max_{a \in \mathcal{A}} \sum_{t=1}^T A_h^{\boldsymbol{\pi}_t, \boldsymbol{\mathcal{R}}_t}(s,a) \leqslant \frac{\ln A}{\eta} + \sum_{t=1}^T \sum_{a \in \mathcal{A}} \pi_{t+1,h}(a|s)\, A_h^{\boldsymbol{\pi}_t, \boldsymbol{\mathcal{R}}_t}(s,a)\,. \tag{19}$$

We fix a comparator policy $\boldsymbol{\pi}$. The combination of the obtained inequality (19) with the application (8) of the performance difference lemma yields

$$\sum_{t=1}^T \left( V_1^{\boldsymbol{\pi}, \boldsymbol{\mathcal{R}}_t}(s_1) - V_1^{\boldsymbol{\pi}_t, \boldsymbol{\mathcal{R}}_t}(s_1) \right) \leqslant \sum_{h=1}^H \sum_{s \in \mathcal{S}} \mu_h^{s_1, \boldsymbol{\pi}, \boldsymbol{\mathcal{T}}}(s) \max_{a \in \mathcal{A}} \sum_{t=1}^T A_h^{\boldsymbol{\pi}_t, \boldsymbol{\mathcal{R}}_t}(s,a)$$

$$\leqslant \frac{H \ln A}{\eta} + \sum_{t=1}^T \sum_{h=1}^H \underbrace{\sum_{s \in \mathcal{S}} \mu_h^{s_1, \boldsymbol{\pi}, \boldsymbol{\mathcal{T}}}(s) \sum_{a \in \mathcal{A}} \pi_{t+1,h}(a|s)\, A_h^{\boldsymbol{\pi}_t, \boldsymbol{\mathcal{R}}_t}(s,a)}_{\text{to be bounded}}\,. \tag{20}$$

We fix $t \in [T]$ and $h \in [H]$ and define a new one-shot policy $\tilde{\boldsymbol{\pi}}_{t+1}^h = \left( \tilde{\pi}_{t+1,h'}^h \right)_{h' \in [H]}$ as follows:

$$\tilde{\pi}_{t+1,h'}^h = \begin{cases} \pi_{h'} & \text{if } h' \leqslant h-1, \\ \pi_{t+1,h'} & \text{if } h' \geqslant h. \end{cases}$$

As $\boldsymbol{\pi}$ and $\tilde{\boldsymbol{\pi}}_{t+1}^h$ coincide in the first $h-1$ stages, we have $\mu_h^{s_1, \boldsymbol{\pi}, \boldsymbol{\mathcal{T}}}(s) = \mu_h^{s_1, \tilde{\boldsymbol{\pi}}_{t+1}^h, \boldsymbol{\mathcal{T}}}(s)$. In addition, the definition of $\tilde{\boldsymbol{\pi}}_{t+1}^h$, the definition of the strategy, Lemma 2, and the definition of advantage functions entail that for all $s \in \mathcal{S}$ and all $h' \geqslant h$,

$$\sum_{a \in \mathcal{A}} \tilde{\pi}_{t+1,h'}^h(a|s)\, A_{h'}^{\boldsymbol{\pi}_t, \boldsymbol{\mathcal{R}}_t}(s,a) = \sum_{a \in \mathcal{A}} \pi_{t+1,h'}(a|s)\, A_{h'}^{\boldsymbol{\pi}_t, \boldsymbol{\mathcal{R}}_t}(s,a) \geqslant \sum_{a \in \mathcal{A}} \pi_{t,h'}(a|s)\, A_{h'}^{\boldsymbol{\pi}_t, \boldsymbol{\mathcal{R}}_t}(s,a) = 0\,.$$

Therefore, the sum marked as "to be bounded" in (20) can be controlled as

$$\sum_{s \in \mathcal{S}} \mu_h^{s_1, \boldsymbol{\pi}, \boldsymbol{\mathcal{T}}}(s) \sum_{a \in \mathcal{A}} \pi_{t+1,h}(a|s)\, A_h^{\boldsymbol{\pi}_t, \boldsymbol{\mathcal{R}}_t}(s,a) = \sum_{s \in \mathcal{S}} \mu_h^{s_1, \tilde{\boldsymbol{\pi}}_{t+1}^h, \boldsymbol{\mathcal{T}}}(s) \sum_{a \in \mathcal{A}} \tilde{\pi}_{t+1,h}^h(a|s)\, A_h^{\boldsymbol{\pi}_t, \boldsymbol{\mathcal{R}}_t}(s,a)$$

$$\leqslant \sum_{h'=h}^H \sum_{s \in \mathcal{S}} \mu_{h'}^{s_1, \tilde{\boldsymbol{\pi}}_{t+1}^h, \boldsymbol{\mathcal{T}}}(s) \sum_{a \in \mathcal{A}} \tilde{\pi}_{t+1,h'}^h(a|s)\, A_{h'}^{\boldsymbol{\pi}_t, \boldsymbol{\mathcal{R}}_t}(s,a)$$

$$= \sum_{s \in \mathcal{S}} \mu_h^{s_1, \boldsymbol{\pi}, \boldsymbol{\mathcal{T}}}(s) \left( V_h^{\tilde{\boldsymbol{\pi}}_{t+1}^h, \boldsymbol{\mathcal{R}}_t}(s) - V_h^{\boldsymbol{\pi}_t, \boldsymbol{\mathcal{R}}_t}(s) \right), \tag{21}$$

where the final equality (21) follows from the equality of distributions $\mu_h^{s_1, \boldsymbol{\pi}, \boldsymbol{\mathcal{T}}} = \mu_h^{s_1, \tilde{\boldsymbol{\pi}}_{t+1}^h, \boldsymbol{\mathcal{T}}}$ at stage $h$ (which holds because $\boldsymbol{\pi}$ and $\tilde{\boldsymbol{\pi}}_{t+1}^h$ coincide in the first $h-1$ stages) together with an application of the performance difference lemma (Lemma 1).

As $\tilde{\boldsymbol{\pi}}_{t+1}^h$ and $\boldsymbol{\pi}_{t+1}$ coincide in the last $h$ stages, we have $V_h^{\tilde{\boldsymbol{\pi}}_{t+1}^h, \boldsymbol{\mathcal{R}}_t}(s) = V_h^{\boldsymbol{\pi}_{t+1}, \boldsymbol{\mathcal{R}}_t}(s)$ for all $s \in \mathcal{S}$. This observation, combined with (21), entails

$$\sum_{s \in \mathcal{S}} \mu_h^{s_1, \boldsymbol{\pi}, \boldsymbol{\mathcal{T}}}(s) \sum_{a \in \mathcal{A}} \pi_{t+1,h}(a|s)\, A_h^{\boldsymbol{\pi}_t, \boldsymbol{\mathcal{R}}_t}(s,a) \leqslant \sum_{s \in \mathcal{S}} \mu_h^{s_1, \boldsymbol{\pi}, \boldsymbol{\mathcal{T}}}(s) \left( V_h^{\boldsymbol{\pi}_{t+1}, \boldsymbol{\mathcal{R}}_t}(s) - V_h^{\boldsymbol{\pi}_t, \boldsymbol{\mathcal{R}}_t}(s) \right),$$

and we thus get, after substitution into (20),

$$\sum_{t=1}^{T}\left(V_1^{\pi,\mathcal{R}_t}(s_1) - V_1^{\pi_t,\mathcal{R}_t}(s_1)\right) \leqslant \frac{H\ln A}{\eta} + \sum_{h=1}^{H}\sum_{s\in\mathcal{S}}\mu_h^{s_1,\pi,\mathcal{T}}(s)\sum_{t=1}^{T}\left(V_h^{\pi_{t+1},\mathcal{R}_t}(s) - V_h^{\pi_t,\mathcal{R}_t}(s)\right). \qquad (22)$$

We obtain telescoping sums on regimes of payoffs. More precisely, with the notation (12),

$$\forall k\in\{2,\dots,R+1\}, \qquad \sum_{t=\tau_{k-1}}^{\tau_k-1}\left(V_h^{\pi_{t+1},\mathcal{R}_t}(s) - V_h^{\pi_t,\mathcal{R}_t}(s)\right) = V_h^{\pi_{\tau_k},\mathcal{R}_{\tau_{k-1}}}(s) - V_h^{\pi_{\tau_{k-1}},\mathcal{R}_{\tau_{k-1}}}(s) \leqslant H-h+1\,,$$

where the upper bound follows from the boundedness of rewards in $[0,1]$. Together with (22), we finally obtain

$$\sum_{t=1}^{T}\left(V_1^{\pi,\mathcal{R}_t}(s_1) - V_1^{\pi_t,\mathcal{R}_t}(s_1)\right) \leqslant \frac{H\ln A}{\eta} + \sum_{h=1}^{H}\sum_{s\in\mathcal{S}}\mu_h^{s_1,\pi,\mathcal{T}}(s)\sum_{k=2}^{R+1}(H-h+1) = \frac{H\ln A}{\eta} + \frac{RH(H+1)}{2}\,,$$

which leads to the claimed regret upper bound after taking the maximum over all policies $\pi$. $\qquad\square$

**Remark 6.** *The arguments between* (20) *and* (22) *may be bypassed in the discounted setting with discount factor $\gamma$; see Agarwal et al. (2021, Section 5.3). More precisely, (with obvious notation, for value functions defined in the standard way for discounted rewards, and for a constant reward function), for each $s\in\mathcal{S}$,*

$$\max_{a\in\mathcal{A}}\sum_{t=1}^{T}A^{\pi_t}(s,a) \leqslant \frac{\ln A}{\eta} + \sum_{t=1}^{T}\overbrace{\sum_{a\in\mathcal{A}}\pi_{t+1}(a|s)\,A^{\pi_t}(s,a)}^{\geqslant 0}$$

$$\leqslant \frac{\ln A}{\eta} + \sum_{t=1}^{T}\underbrace{\frac{1}{1-\gamma}\sum_{s'\in\mathcal{S}}\mu^{s,\pi_{t+1}}(s')\sum_{a\in\mathcal{A}}\pi_{t+1}(a|s')\,A^{\pi_t}(s',a)}_{=V^{\pi_{t+1}}(s)-V^{\pi_t}(s)} = \frac{\ln A}{\eta} + \underbrace{V^{\pi_{T+1}}(s) - V^{\pi_1}(s)}_{\leqslant 1/(1-\gamma)}\,,$$

*where the first inequality is by Lemma 3, where the non-negativity is guaranteed by monotonicity of weights (see Lemma 2), where the second inequality comes from the fact that distributions induced by a starting state $s$, a given policy, and a given transition function put a probability mass at least $1-\gamma$ on $s$, no matter the policy and transition function (this is the property extremely specific to the discounted setting), where the equality to $V^{\pi_{t+1}}(s) - V^{\pi_t}(s)$ is by the performance difference lemma, and where the final equality is by telescoping. The inequality obtained above is the key; the rest of the proof merely consists of yet another (now standard) application of the performance difference lemma:*

$$\sum_{t=1}^{T}\left(V^{\pi}(s_1) - V^{\pi_t}(s_1)\right) = \sum_{t=1}^{T}\frac{1}{1-\gamma}\sum_{s\in\mathcal{S}}\mu^{s_1,\pi}(s)\sum_{a\in\mathcal{A}}\pi(a|s)\,A^{\pi_t}(s,a) \leqslant \frac{1}{1-\gamma}\sum_{s\in\mathcal{S}}\mu^{s_1,\pi}(s)\underbrace{\max_{a\in\mathcal{A}}\sum_{t=1}^{T}A^{\pi_t}(s,a)}_{\leqslant(\ln A)/\eta+1/(1-\gamma)}\,,$$

*which is the bound claimed by Agarwal et al. (2021, Section 5.3).*

We conclude this section with a proof of Lemma 3.

*Proof of Lemma 3.* First, a bound "à la Pisier" yields that for all sequences of payoffs $g_{t,j}$, possibly signed and unbounded:

$$\max_{k\in[K]}\sum_{t=1}^{T}g_{t,k} = \frac{1}{\eta}\ln\left(\max_{j\in[K]}\exp\left(\eta\sum_{t=1}^{T}g_{t,j}\right)\right)$$

$$\leqslant \frac{1}{\eta}\ln\left(\sum_{j\in[K]}\exp\left(\eta\sum_{t=1}^{T}g_{t,j}\right)\right) = \frac{\ln K}{\eta} + \frac{1}{\eta}\sum_{t=1}^{T}\ln\left(\sum_{j\in[K]}w_{t,j}\exp(\eta g_{t,j})\right),$$

where the equality follows by telescoping: indeed, by definition of the weights,

$$\sum_{j\in[K]} \underbrace{\exp\left(\eta\sum_{t=1}^{T} g_{t,j}\right)}_{=v_{T+1,j}} = K\prod_{t=1}^{T} \frac{\sum\limits_{j\in[K]} v_{t+1,j}}{\sum\limits_{j\in[K]} v_{t,j}} = K\prod_{t=1}^{T} \frac{\sum\limits_{j\in[K]} v_{t,j}\exp(\eta g_{t,j})}{\sum\limits_{j\in[K]} v_{t,j}} = K\prod_{t=1}^{T} w_{t,j}\exp(\eta g_{t,j}).$$

Second, by the application of Jensen's inequality to the convex function $x\mapsto x\ln x$,

$$\left(\sum_{j\in[K]} w_{t,j}\exp(\eta g_{t,j})\right)\ln\left(\sum_{j\in[K]} w_{t,j}\exp(\eta g_{t,j})\right) \leqslant \sum_{j\in[K]} w_{t,j}\exp(\eta g_{t,j})\ln\big(\exp(\eta g_{t,j})\big),$$

that is, after rearranging and given the definition of the weights $w_{j,t+1}$,

$$\ln\left(\sum_{j\in[K]} w_{t,j}\exp(\eta g_{t,j})\right) \leqslant \eta\sum_{j\in[K]} w_{t+1,j}\, g_{t,j}.$$

The claimed bound follows from combining the two inequalities obtained. $\qquad\square$

## B  Proof of the performance difference lemma

One of the first references stating the performance difference lemma (in the discounted setting) is Kakade & Langford (2002). Statements (possibly of generalizations) of this lemma for $H$–episodic MDPs are ubiquitous in the literature (see, e.g., Cai et al., 2020, Lemma 3.2 for a simple statement, and Shani et al., 2020, Lemma 1 for an extension to approximated advantage functions). We state yet another, straightforward, generalization, in terms of advantage and value functions starting at a given stage $h$; this generalization is useful in the proof of Theorem 4 in Appendix A.

**Lemma 1** (Performance difference lemma)**.** *Let $\mu_{h'}^{s_1,\boldsymbol{\pi},\mathcal{T}}$ be the distribution of the state $s_{h'}$ of the $h'$–th stage, starting from the state $s_1$ in the first stage, following the stationary policy $\boldsymbol{\pi}$ and the transition kernels $\mathcal{T}$. In a MDP with transition kernels $\mathcal{T}$, for all pairs $\boldsymbol{\pi},\boldsymbol{\pi}'$ of stationary policies, for all reward functions $\mathcal{R}$, and for all stages $h\in[H]$,*

$$\sum_{s\in\mathcal{S}} \mu_h^{s_1,\boldsymbol{\pi},\mathcal{T}}(s)\Big(V_h^{\boldsymbol{\pi},\mathcal{R}}(s) - V_h^{\boldsymbol{\pi}',\mathcal{R}}(s)\Big) = \sum_{h'=h}^{H}\sum_{s\in\mathcal{S}} \mu_{h'}^{s_1,\boldsymbol{\pi},\mathcal{T}}(s)\sum_{a\in\mathcal{A}} \pi_{h'}(a|s)\, A_{h'}^{\boldsymbol{\pi}',\mathcal{R}}(s,a)\,.$$

*In particular, for $h=1$,*

$$V_1^{\boldsymbol{\pi},\mathcal{R}}(s_1) - V_1^{\boldsymbol{\pi}',\mathcal{R}}(s_1) = \sum_{h'=1}^{H}\sum_{s\in\mathcal{S}} \mu_{h'}^{s_1,\boldsymbol{\pi},\mathcal{T}}(s)\sum_{a\in\mathcal{A}} \pi_{h'}(a|s)\, A_{h'}^{\boldsymbol{\pi}',\mathcal{R}}(s,a)\,.$$

*Proof.* We denote by $\mathbb{P}^{s_1,\boldsymbol{\pi},\mathcal{T}}$ the probability distribution underlying the $H$–episodic MDP $(s_1,a_1,\ldots,s_H,a_H)$ starting at $s_1$, drawing actions according to $\boldsymbol{\pi}$, and subject to the transition kernels $\mathcal{T}$. In particular, by definition, for any function $f:\mathcal{S}\times\mathcal{A}\to\mathbb{R}$ and all $h'\in[H]$,

$$\sum_{s\in\mathcal{S}} \mu_{h'}^{s_1,\boldsymbol{\pi},\mathcal{T}}(s)\sum_{a\in\mathcal{A}} \pi_{h'}(a|s)\, f(s,a) = \mathbb{E}^{s_1,\boldsymbol{\pi},\mathcal{T}}\big[f(s_{h'},a_{h'})\big]\,.$$

Letting successively $f$ be $A_{h'}^{\boldsymbol{\pi}',\mathcal{R}}$ for $h\leqslant h'\leqslant H$ and using the definition $A_{h'}^{\boldsymbol{\pi}',\mathcal{R}} = Q_{h'}^{\boldsymbol{\pi}',\mathcal{R}} - V_{h'}^{\boldsymbol{\pi}',\mathcal{R}}$,

$$\sum_{h'=h}^{H}\sum_{s\in\mathcal{S}} \mu_{h'}^{s_1,\boldsymbol{\pi},\mathcal{T}}(s)\sum_{a\in\mathcal{A}} \pi_{h'}(a|s)\, A_{h'}^{\boldsymbol{\pi}',\mathcal{R}}(s,a) = \mathbb{E}^{s_1,\boldsymbol{\pi},\mathcal{T}}\left[\sum_{h'=h}^{H}\big(Q_{h'}^{\boldsymbol{\pi}',\mathcal{R}}(s_{h'},a_{h'}) - V_{h'}^{\boldsymbol{\pi}',\mathcal{R}}(s_{h'})\big)\right]. \qquad (23)$$

Now, by definition of the $Q$–values, recalling that $\boldsymbol{r}$ denotes the mean-payoff functions associated with $\mathcal{R}$, we have, for $h' \leqslant H - 1$,

$$\forall (s, a) \in \mathcal{S} \times \mathcal{A}, \qquad Q_{h'}^{\boldsymbol{\pi}', \mathcal{R}}(s, a) = r_{h'}(s, a) + \sum_{s' \in \mathcal{S}} \mathcal{T}_{h'}(s' \mid s, a)\, V_{h'+1}^{\boldsymbol{\pi}', \mathcal{R}}(s'). \qquad (24)$$

By definition of the MDP, for any function $g : \mathcal{S} \to \mathbb{R}$,

$$\mathbb{E}^{s_1, \boldsymbol{\pi}, \mathcal{T}}\left[\sum_{s' \in \mathcal{S}} \mathcal{T}_{h'}(s' \mid s_{h'}, a_{h'})\, g(s')\right] = \mathbb{E}^{s_1, \boldsymbol{\pi}, \mathcal{T}}\left[g(s_{h'+1})\right].$$

Thus, letting $s = s_{h'}$ and $a = a_{h'}$ in (24) and taking expectations yields

$$\mathbb{E}^{s_1, \boldsymbol{\pi}, \mathcal{T}}\left[Q_{h'}^{\boldsymbol{\pi}', \mathcal{R}}(s_{h'}, a_{h'})\right] = \mathbb{E}^{s_1, \boldsymbol{\pi}, \mathcal{T}}\left[r_{h'}(s_{h'}, a_{h'})\right] + \mathbb{E}^{s_1, \boldsymbol{\pi}, \mathcal{T}}\left[V_{h'+1}^{\boldsymbol{\pi}', \mathcal{R}}(s_{h'+1})\right].$$

For $h' = H$, we have $Q_H^{\boldsymbol{\pi}', \mathcal{R}}(s, a) = r_H(s, a)$. As a consequence of the equalities above, a telescoping sum appears in the right-hand side of (23):

$$\mathbb{E}^{s_1, \boldsymbol{\pi}, \mathcal{T}}\left[\sum_{h'=h}^{H} \left(Q_{h'}^{\boldsymbol{\pi}', \mathcal{R}}(s_{h'}, a_{h'}) - V_{h'}^{\boldsymbol{\pi}', \mathcal{R}}(s_{h'})\right)\right]$$

$$= \mathbb{E}^{s_1, \boldsymbol{\pi}, \mathcal{T}}\left[r_{h'}(s_H, a_H) + \sum_{h'=h}^{H-1} \left(r_{h'}(s_{h'}, a_{h'}) + V_{h'+1}^{\boldsymbol{\pi}', \mathcal{R}}(s_{h'+1})\right) - \sum_{h'=h}^{H} V_{h'}^{\boldsymbol{\pi}', \mathcal{R}}(s_{h'})\right]$$

$$= \mathbb{E}^{s_1, \boldsymbol{\pi}, \mathcal{T}}\left[\sum_{h'=h}^{H} r_{h'}(s_{h'}, a_{h'})\right] - \mathbb{E}^{s_1, \boldsymbol{\pi}, \mathcal{T}}\left[V_1^{\boldsymbol{\pi}', \mathcal{R}}(s_h)\right].$$

Finally, the tower rule shows that

$$\mathbb{E}^{s_1, \boldsymbol{\pi}, \mathcal{T}}\left[\sum_{h'=h}^{H} r_{h'}(s_{h'}, a_{h'})\right] = \mathbb{E}^{s_1, \boldsymbol{\pi}, \mathcal{T}}\left[V_1^{\boldsymbol{\pi}, \mathcal{R}}(s_h)\right].$$

The proof is concluded by collecting all the bounds. $\qquad\square$

