# OpenReview forum: "Policy Optimization via Adv2: Adversarial Learning on Advantage Functions"
_TMLR — Accepted by TMLR_

### Review · Reviewer_YrCU · 2025-02-23

**Summary Of Contributions:**

This paper provides a comprehensive study on reducing learning adversarial Markov decision processes (MDPs) to adversarial learning, emphasizing generality beyond conventional approaches. The authors extend previous work by demonstrating that this reduction is not limited to exponential weights strategies but can accommodate a broad family of adversarial learning methods, including those based on advantage functions rather than Q-values. These extensions enable broader applicability and improved theoretical performance.

**Audience:**

Yes

**Claims And Evidence:**

Yes

**Requested Changes:**

Please see the above.

**Strengths And Weaknesses:**

Adversarial MDPs are a crucial component of RL theory, and many existing algorithms have leveraged techniques from adversarial learning to address this problem. This work formally establishes the mathematical consistency between adversarial MDPs and adversarial learning, effectively bridging these two important areas of learning theory. Additionally, it incorporates various techniques from adversarial learning to enhance the theoretical performance and applicability of adversarial MDP methods.

I do not see any major shortcomings in this work. However, in existing literature, it is common to compare methods through empirical evaluations. While not strictly necessary, incorporating some experimental validation would further strengthen the impact of the results. That being said, I highly appreciate the contribution this paper makes in bridging adversarial MDPs and adversarial learning, providing a formal and insightful connection between these two important areas of learning theory.

---

> ### Author Response · Authors · 2025-03-06
> **Answer to Reviewer YrCU**
>
> We thank the reviewer for this summary, which we fully share, including the limitation pointed: the lack of experimental validation. Our comments on this are to be found in the 'General answer' above.

---

### Review · Reviewer_2ihs · 2025-03-02

**Summary Of Contributions:**

This paper studies episodic MDPs where an adversary chooses a reward function at each step and episode. In contrast, transition kernels stay the same across episodes (although they may vary within one episode).

The authors describe adversarial strategies leading to sublinear regret bounds, then introduce a new sequential strategy based on estimated advantage functions. When the strategy is a Boltzmann distribution weighted by advantage functions, adversarial regret bounds become tighter than previous works.

Stronger notions of regret are then analyzed: strongly adaptive and tracking regret. The authors show that in both cases, their advantage-based strategy leads to sublinear regret.

Finally, this notion of adversarial regret is studied in imitation learning, where similar bounds can be deduced from properly defined strategies.

**Audience:**

No

**Claims And Evidence:**

No

**Requested Changes:**

I encourage the authors to perform a deep review of their work and clarify the following points:
- what problem does this work solve that previous ones did not
- what technical novelties are made to solve the aforementioned problem?
- what are the shortcomings/limitations of the proposed method?

The authors should also rearrange the sections: one section - one idea. I think this will help in clarifying the authors' intents.
- Introduction:
-- Describe the context. What is an adversarial MDP? What is an adversarial learning strategy? How is it different from "standard" policy optimization?
-- Shortly explain what's missing in previous work and why it is important to fill this gap.
-- Shortly list the contributions (sec. 1.2 is too long). On a high level, what does this work bring?

- Related work:
-- it should be better distinguished with the introduction, perhaps make it a section on its own.

- Preliminaries:
-- All notations, methods from previous works, and background concepts should appear in one place. This will greatly help identify the contributions of this work. These are currently spread across Sec. 2, 2.1, 2.2, 3, and 3.1.

- Contributions

- Discussion: Another missing piece of this work is a discussion/conclusion. What is left to be done that this work hasn't addressed?

Please avoid referring back and forth to other sections if not needed. Explain your idea instead.

Notations are sometimes confusing:
- the mean payoff is denoted by $r_{t,h}(s,a)$ in Sec 2, §3, whereas in §5, it becomes a random variable $r_{t,h}\sim\mathcal{R}_{t}.$
- Policies are sometimes stationary, sometimes not. In particular, the objective in Eq. 2 is unclear: Why maximize over stationary policies only? If the reward is adversarial and changes over time, can't $R_T$ be negative? Are $\mathbf{\pi}$ and $\mathbf{\pi}_t$ related? Or is $\mathbf{\pi}$ an "absolute" policy based on the inherent setup? From eq 3 to 4, why do policies become non-stationary?
- In Definition 1, why are reward vectors in $[-M,M]^K$ and not in $[0,1]^K$ as suggested by the reward domain definition in Sec 2, §2?
- In Definition 1, what is $w_t$? How should this bound be interpreted? What is $\varphi_t$?

**Strengths And Weaknesses:**

The current version has flaws in several aspects, which I detail in increasing order of importance.

**Clarity: ** The structure of this work is hard to parse. Comparison with related works appears in the introduction, then again in Sec. 7 and others. Background notations and methods are all over. In terms of content, the study needs to be contextualized as soon as possible in the paper (abstract + intro). I cannot figure out the goal of this work, partly because I am unfamiliar with the notion of sequential strategies: How are they different from policy optimization? What is or is not a potential function? Why/when do they provide more practical methods for RL than other tools? Additionally, the way cross references to other works are used makes reading hard: these are often used to replace an explanation. The paper should be more self-contained, with all required notions explicitly described.

**Motivation: ** I do not understand what problem this paper is trying to solve. Given that different notions of regret can be considered, what is a "good" criterion for adversarial learning?  What is the gap in previous work that this work tries to fill?

**Novelty: ** I did not thoroughly go through the proofs, but all derivations seem to be based on one trick: take a strategy that depends on advantage functions. From thereon, regret bounds are more or less straightforwardly deduced from previous results (PD lemma). In other words, what are the technical contributions of this work? What are its technical challenges besides estimating the advantage function? On the other hand, I do not see any algorithmic contributions that may be tested on adversarial learning benchmarks.

---

> ### Author Response · Authors · 2025-03-06
> **Answer to Reviewer 2ihs, part 1**
>
> We thank the reviewer for the comments, that will be helpful to us to better state our goals and to better define the audience targeted when revising our article.
>
> &nbsp;
>
> **On the main grounds regarding clarity / motivation / novelty:**
>
> **1.**
> Indeed, this article is not trying to solve a specific model or to fill a gap in previous work. Its focus would rather be a review of, and systematic improvement of, the connections between (plain) adversarial learning and the study of adversarial MDPs. Reviewer YrCU actually states our goals in a clearer manner than we did: 'This work formally establishes the mathematical consistency between adversarial MDPs and adversarial learning, effectively bridging these two important areas of learning theory.'
>
> **2.**
> Because of that, we agree that the audience intended is formed by researchers that are already aware of the (relatively recent) topic of adversarial learning for MDPs, and even, policy optimization for adversarial learning for MDPs --- i.e., that are aware of, e.g., the works by Shani et al. (2020) and Cai et al. (2020). We also implicitely assumed that such researchers would have been exposed to plain adversarial learning --- a topic that emerged with an article by Littlestone and Warmuth (1989) and has been an active domain of research since then, with a landmark monograph by Cesa-Bianchi and Lugosi (2006) summarizing the fundamental results of this domain. In particular, this monograph defines plain adversarial learning and defines potential-based strategies in the first section of its first chapter (Section 2.1, pages 9--15). Because of the existence of this monograph, which we refer to right before Section 3.1, we kept our reminder of plain adversarial learning to the minimum --- Section 3.1 mainly recalls standard notation and strategies for plain adversarial learning but does not even try to be a primer on plain adversarial learning. We did so to avoid artificially increasing the length of our submission and to go faster to the point where we wanted to start from, which is the derivation by Shani et al. (2020, Section 6) recalled in Section 3.2. We would of course happily revise this editorial choice if requested to do so from the reviewers and the action editor.
>
> **3.**
> The main technical contributions are in Sections 4, 5, and 8 (the three section titled 'Extension $n$: [...]', with $n \in \{1,2,3\}$). Section 4 extends the convergence-of-the-last-iteration results from exponential weights to a much wider class of plain-adversarial learning strategies --- an extension that Reviewer 1AvV specifically underlines, as the class of strategies considered may have impacts beyond the specific focus of this article. This would be the main technical contribution. Section 5 introduces more complex notions of regrets in adversarial MDPs --- and basically explains that more complex notions of regret in plain adversarial learning may be transferred to regret for MDPs. This observation is simply pushing the logic of the reduction discussed in Section 3 and may have consequences and impacts on future works in adversarial learning for MDPs. Admittedly, this contribution is rather intuitive once the reduction of Section 3.2 was clearly stated; stating the reduction of Section 3.2 is therefore the main non-technical contribution of this article. Finally, Section 8 revisits the research topic called imitation learning and puts it under the umbrella of plain adversarial learning (which is also referred to as aggregation of experts). This Section 8 (see Remark 4) actually questions the pertinence of a line of research in imitation learning and provides simpler and more transparent approach to address this question.
>
> &nbsp;
>
> **As a consequence, on the changes to be performed:**
>
> **4.**
> As written in the 'General Answer' above, we agree that the article lacks a conclusion and the indication of future resarch avenues. As answered to Reviewer 1AvV, we will state more clearly that our study relies on a value-function oracle and assumes full monitoring. We also commit to clarify the audience targeted (see 2. above) and the focus of the article (see 1. above). However, based on the feedback from the two other reviewers and our own intuitions, we would stick to the current exposition which, in our opinion, already follows the principle of 'one idea - one section'. It is true that this leads to introducing notation and background concepts throughout the text (even in Section 8!), but we believe that it is more efficient to do so when needed, rather than inflicting several pages of notation and concepts before any formal result. We state the first theorem at the end of page 6 and would rather not postpone it.

---

> > ### Author Response · Authors · 2025-03-06
> > **Answer to Reviewer 2ihs, part 2**
> >
> > **On minor comments and remarks:**
> > - First notation issue: Because of the definition of the regret and thanks to the tower rule, we never use the actual rewards in our proofs, and only need to refer to mean payoffs; we had overloaded the notation, as also underlined by Reviewer 1AvV, and agree that this was a bad idea; we will fix the issue and use two different pieces of notation
> > - The questions about the very definition of regret for adversarial MDPs are actually deep questions. The classic definition of regret in adversarial MDPs is not ours, we copied it from the literature. We agree that comparing the performance of a sequence of policies built over time to a single stationary policy $\pi$ (which is indeed an 'absolute' policy) is a limited comparison, but, for various reasons which we did not summarize in this article, this is what is classically done. Yes, the regret could be negative (as, by the way, in plain adversarial learning) but the aim is only to bound it from above. That being said, exactly because of the limited comparison mentioned, one of our goals was to introduce more complex notions of regrets for adversarial MDPs! On a side note: yes, an indexation in $t$ appears between (3) and (4) but there is no real reason for doing so, we will take care of this when revising the article.
> > - Definition 1: Yes, rewards fed to the MDPs are $[0,1]$--valued, but the reward vectors to be considered for plain-adversarial learning strategies are formed by advantage functions, which takes values in subsets of $[-H,H]$. See the comments before Equation (6): we indicate therein what $M_h$ we take and why.
> > - The $w_t$ in Definition 1 were defined right above: these are the convex weights output by the plain-adversarial learning strategies. (Here again, we acknowledge that the exposition is intended for readers already familiar with plain adversarial learning, and that our exposition merely introduces notation but does not try to explain the background concepts.) The functions $\varphi_t$ are the mathematical formulation of a strategy: a mapping from information available at the beginning of round $t$ to the set of all convex weights. There is a typo here that prevented the reviewer from understanding the definition, and we apologize for that: we wanted to write $\varphi_t : \mathbb{R}^{K(t-1)} \to \mathcal{P}([K])$. We will fix the issue when revising the article.

---

### Review · Reviewer_1AvV · 2025-03-04

**Summary Of Contributions:**

This paper studies episodic reinforcement learning in the regret setting in adversarial MDPs, namely episodic MDPs with stochastic transition function but with reward functions that are arbitrary as if they were generated by an oblivious adversary. The paper revisits algorithmic approaches that fall under the umbrella of policy optimization. Such approaches in the literature mostly build on the celebrated exponential weights strategy that rendered efficient to yield order-optimal regret guarantees. Focusing on the full information setting, the paper aims to offer a unified view of the approaches presented in the literature that is presented under the name Adv2. This unification results in, among others, sharper regret guarantees (and convergence of last iterate) in some cases and extending some existing results to the case of more refined regret criteria – such as adaptive regret and tracking regret. The larger part of the paper assumes that value functions are known and focuses on how to tackle the adversarial nature of rewards. Finally, a variant of Adv2 is discussed and analyzed for the case of aggregation strategies.

**Audience:**

Yes

**Broader Impact Concerns:**

I believe there is no concern associated to the developments and results reported in the paper.

**Claims And Evidence:**

Yes

**Requested Changes:**

See above.

**Strengths And Weaknesses:**

This work investigates policy optimization in episodic MDPs, which is related to an interesting avenue of research in theoretical RL. The focus is on adversarially generated reward functions in the full information setting. I personally find such reward models quite relevant in (theoretical) RL but also believe that it could impact the practice in its applied side, too. (The authors did not discuss such things perhaps due to space limitation, which is fine. Nevertheless, I believe providing some brief pointers to such relevance could be useful for general audience.)

The paper is written very well, admits a clear organization and presentation, and is overall a nice read. It offers a rich literature review, carefully positioning its contributions to the relevant work and citing most relevant papers I am aware of. Also, short contribution summaries at the beginning of various sections appeared quite helpful. I must mention, however, that it lacks an epilogue (e.g., a ‘conclusion’ section) that could sketch some future directions to continue this research.

Regarding technical aspects: the paper revisits policy optimization in adversarial MDPs (in the full information setting), focusing on works that build on adversarial learning strategies. While providing a technical overview of existing work, it does a good a job in identifying and proposing a systematic approach that encompasses a much larger class of strategies. Overall I believe it is a nice contribution although at first sight it may render incremental. The authors propose a number of improvements and extensions to the state-of-the-art which turns out to happen thanks to the modular form of the proposed Adv2 strategy. Especially, introducing and identifying strategies satisfying the weight monocity property is, to me, a novel contribution.

My major concern regarding the results is that the larger part of the paper assumes that value functions are known, shifting the focus onto tackling the adversarial nature of rewards. Of course, the authors eventually turn to the so-called ``practical case'' where the transition function is unknown (and hence value function) and explain how the regret under this case related to the regret with regards to the value estimates. But then it is assumed the algorithm would work with value estimates instead of true values. Put differently, the preceding sections were written as if there is a “value oracle” and Section 7 tries to account for lacking such an oracle. Arguably, much of technical hurdles in RL algorithms arise due to the complex nature of noise in value estimates (using direct or indirect methods), and the wording of the section sounds to me like an oversimplification of problem. Thus, my concern is that the claims of Section 7 could be perceived quite differently by different readers, and I am afraid the current execution of this very section could be fully misleading to some.

Overall, excluding its Section 7, the paper may be nicely and consistently read while keeping in mind the existence of some “value oracle” throughout. That said, I do not really see what values the results/discussions in Section 7 may bring in, other than interrupting the previous flow. An alternative execution could treat the discussion in Section 7 for warming up future research which could investigate this in further depth.

Another major concern is that the introduction and abstract do not make it clear that the paper considers full information setting and (mostly) assumes known values – unless I am missing something. This must be fixed accordingly.

Other comments are listed below:

- Regarding the sentence “the upper bound of Thm. 4 is smaller than a constant when the reward functions do not vary over time” (in p. 12), could you please elaborate whether this is of relevance when value functions must be estimated?

- After Corollary 2, you conclude that $\eta$ must be as large as possible. However, this could come at some sacrifice in other performance aspects. Is it correct?

- Some discussions in the proof of Lemma 2 – e.g., on top of p. 20 – are worthwhile to be included in the main text, considering the novelty of “weight monotonicity”.

- p. 3: $r_{h,t}$ is used to denote both the mean reward and realization at stage $h$ of episode $t$. It would be better to avoid such a notation overloading.

- Beginning of Section 3.1: $K$ is not yet introduced.

- $\varphi_t: \mathbb R^{K(t-1)}$ ==> It looks like that something is missing (e.g., a co-domain or range)

Potential Typos/Grammar Mistakes:

- p. 1: by by ==> by

- p. 2 (and elsewhere): policies output ==> It appears to me that "output policies" is more accurate.

- p. 2: for instance derived, in ==> for instance, derived in

- p. 6: Euclidean norm ==> the Euclidean norm

- p. 7: relies in ==> Did you mean 'lies in' or 'relies on'

- p. 8: a possibly a lower ==> a possibly lower

- p. 9: Definition 1 regret ==> To me, 'regret' is redundant

- p. 9 (and elsewhere): ... replace ... by ==> replace ... with ---- Of course in the passive form, it would be '[to be] replaced ... by'

- p. 10 (and elsewhere): subintervals ==> sub-interval (to be consistent with earlier uses)

- p. 11: We consider sequences ==> Shouldn't it be sequence? Or did you mean for various $T$?

- p. 12: more complex that ==> ... than

- p. 12: regimes shifts ==> regime shifts

- p. 12: $1\sqrt{T}$ ==> $1/\sqrt{T}$

- p. 14: epochs ==> Did you mean 'episodes'?

- p. 20: The calculation above show ==> ... shows

---

> ### Author Response · Authors · 2025-03-06
> **Answer to Reviewer 1AvV**
>
> We agree with all the positive comments, as well as the limitations underlined. The main concerns (e.g., lack of a conclusion, oversimplification in Section 7) and suggestion (alternative execution of Section 7 as warming up future research) are answered in the 'General Answer' above.
>
> Together with the rewriting suggested therein, we commit to make clearer in our revision that our results are based on a value oracle and under a full-information monitoring (in the abstract + introduction, but also throughout the text). Yet, our results, even under these restrictions, may turn out to be useful in the future: for instance, for strategies that would proceed epoch-wise, by estimating the environment on past epochs and using some estimate of the transition kernel throughout the current epoch (as in, by the way, Tiapkin et al., 2024). Then, it's as if the environment was known, and the value oracle corresponds to merely computing the value function (e.g., through dynamic programming). We will go over this question of a value oracle more explicitely, and at several places in the article (e.g., in the introduction as well as in the reshuffled Section 7).
>
> Answers to minor comments
> - Regarding the sentence 'The bound of Theorem 4 has a smaller order of magnitude the one of Theorem 1': we will indeed comment that this bound (and the corresponding bound without shifts in Agarwal et al., 2021, Section 5.3) may be spoiled by errors linked to estimating transition kernels
> - Comments on $\eta$ after Corollary 2: Note that the same issue arises for the bound of Agarwal et al. (2021, Section 5.3).
> In principle and intuitively, there should be a trade-off impeding to arbitrarily increase $\eta$
>  While we have not identified any theoretical barriers for our setting, we suspect that practical performance may be adversely affected. We will recommend in the new conclusion section that empirical evaluations be conducted to assess the impact of varying $\eta$.
> - Some discussions in the proof of Lemma 2 to be put in the main text: We will happily follow this suggestion.
> - Notation overloading: we agree that this is an issue and will fix it (actually, the true rewards are never referred to, as, by the tower rule, all regret proofs directly involve mean-payoff functions)
> - $K$ not introduced in the beginning of Section 3.1: indeed, we will start the section with a sentence explaining what $K$ is in the expert setting.
> - Learning strategies: yes, the range was missing, we meant $\varphi_t : \mathbb{R}^{K(t-1)} \to \mathcal{P}([K])$
> - Potential Typos/Grammar Mistakes: We agree with all of them and will fix them.

---

### Author Response · Authors · 2025-03-06
**General answer to all reviews**

We agree with all comments from Reviewers 1AvV and YrCU, both the positives ones and the limitations pointed out.

&nbsp;

**The main positive comments we read are:**
- The summary of our goal: 'This work formally establishes the mathematical consistency between adversarial MDPs and adversarial learning, effectively bridging these two important areas of learning theory.' (Reviewer YrCU)
- Our wish to offer a rich literature review (Reviewer 1AvV)
- The identification of a class of adversarial-learning strategies that is interesting per se, and could have applications beyond this article, namely, strategies satisfying monotonicity of weights (Reviewer 1AvV)
- The comments on the organization, presentation, and clarity, except for Section 7 (Reviewers 1AvV and YrCU)

&nbsp;

**Each of the three reviewers points out one or several limitations of the following list**; these limitations actually go hand in hand as we will explain:
- Issue 1: Section 7 oversimplifies the treatment of the pratical cases where no 'value oracle' is available
- Issue 2: The lack of experimental validation
- Issue 3: The lack of a conclusion and of future research directions

Indeed, virtually all the strategies considered by the references mentioned in Section 7 have a step where policies are obtained through exponential weights on estimated Q-functions, together with many other steps and tricks specific to the settings considered. The point of our submission is exactly that this step of these strategies may be replaced by the application of another adversarial learning strategy, with the (many) other steps of the algorithm remaining unchanged. This would typically not change the associated theoretical guarantees, as only bounds for exponential weights of order $\sqrt{T \ln K}$ were used in these proofs: the part of the proof where this bound is used is the easy part, while the rest of the analysis is usually involved and relies on specific assumptions (e.g., existence of linear representations and their clever use in the strategies). However, this simple modification, while impactless on the theoretical bound, may change dramatically the practical performance obtained by these strategies. (As adversarial learning strategies, we would recommend the adversarial strategies ML-Poly and ML-Prod, and others, which are available through the R package Opera; see https://cran.r-project.org/web/packages/opera.)

In a nutshell, we agree that our current writing for Section 7 does not underline enough that in all references cited, the hard work is everywhere but in the step where exponential weights are taken over estimated Q-functions [see Issue 1]. We will fix that. That being said, an interesting research direction is, in our opinion, to take several of these references, which consider different (theoretical and experimental) scenarios, and test the impact of changing the way the estimated Q-functions are combined to yield the output policies over time. This would require quite some work and would probably take an entire article to report the results. Also, this set of authors is rather theory-oriented. This all explains the lack of an experimental section [see Issue 2]. However, performing this empirical evaluation of the impact of the adversarial strategy used to combine the estimated Q-functions (and also, to rather combine instead estimated advantage functions) looks to us as an interesting research program that would lead to an interesting follow-up article for TMLR: we propose to clearly outline this research program in a conclusion section [see Issue 3].

**As a conclusion:** Given reviewers' feedback (especially the alternative execution suggested by Reviewer 1AvV) and our analysis above, the optimal change would be to move the current Section 7 as a final section and deeply rewrite it, so as to explain how the 'oracle' results described in the earlier sections could possibly impactful, but that this potential impact needs to be evaluated through extensive numerical experiments (relying on the several ones already performed in the mentioned references). We commit to do so.

---

### Author Response · Authors · 2025-03-28
**Revision uploaded**

We just uploaded a revised version of the manuscript --- by implementing all changes promised in this review thread.

Local changes are in blue.

The main change is not in blue and concerns the new Section 8 (numbered Section 7 in the initial submission): therein, we discuss the impacts of our results in the absence of oracles providing value function and state future research directions.

Also: we apologize for the delay taken in submitting this revision.

---

> ### Comment · Reviewer_1AvV · 2025-04-03
> **Feedback on revised paper**
>
> Thanks for the revised version. I believe it addresses my major concerns. The only major comment I may have is that the abstract does not yet reflect that main results are obtained building on a value oracle assumption. Please revise it accordingly.
>
> Some (potential) grammatical mistakes:
>
> - p. 5: with the fundamentals concepts ==> ... fundamental concepts
> - p. 5: this set of options [K] with the set of actions $\mathcal A$ ==> this set [K] of options with the set $\mathcal A$ of actions OR this set of options, [K], with ...
> - p. 17: with no impact of the theoretical guarantees  ==> with no impact on ...
> - p. 18: ... that would defined ==> ... that would define OR that would have defined
> - p. 19: to replace ... by ==> ... with

---

> > ### Author Response · Authors · 2025-04-04
> > **Abstract revised**
> >
> > We thank heartfully the reviewer for the careful verification of our changes.
> >
> > We were focused on the changed in the main body and indeed had totally forgotten to update the abstract, which we just did --- see the revised PDF submitted. (We also corrected all grammatical mistakes raised.) We apologize for the inconvenience.

---

### Author Response · Authors · 2025-05-09
**Revision comments**

We thank the action editor and the reviewers for handling this submission. We uploaded a revision, obtained after a careful pass on the entire text. We performed many small corrections --- in particular (but not only), to address the issues underlined by the action editor. (We created an additional box describing the setting of Section 2.1, because the former Box A needed to stay in Section 7, as its aim is to compare the settings of Sections 2.1 and 7.) We also corrected occasional typos (in particular, in the re-shuffled Section 8; in the statement of Corollary 1, where there was a confusion between the boundaries of the intervals and their lengths; and in Remark 6 where there were normalization issues regarding $1-\gamma$).

---

> ### Author Response · Authors · 2025-05-16
> **Changes performed (in particular, concerning how imitation learning is cited)**
>
> Thanks a lot for your dedication and detailed reading again!
>
> **We submitted an updated version.**
>
> For the first three comments, we corrected into 'Trust Region Policy Optimization', 'which builds on the present work' (we want to mention our precedence), and 'an entropic regularization'.
>
> For the comment on imitation learning: Indeed, while the mentioned references (Cheng et al., 2020; Liu et al., 2023) do state that their work corresponds to imitation learning, this is only one way to address the problem of imitation learning. This is why we
> - dropped all occurrences of imitation learning (including in the abstract) and simply write 'aggregation (or orchestration) of expert policies' instead,
> - except at two places (page 4, pararaph Extension 3 and beginning of Section 7 page 15).
>
> We write, for instance, at the beginning of Section 7: 'The aim is to mimic the performance of the overall best convex combination of expert policies [...], which corresponds to an aggregation (or orchestration) of expert policies. This setting was also termed learning from multiple oracles (which may be understood as a specific paradigm in the vast imitation-learning literature) by Cheng et al. (2020) and Liu et al. (2023).

---

### Decision · Action_Editor_ne2d · 2025-04-17

**Recommendation:** Accept with minor revision

**Comment:**

This paper studies policy optimization in adversarial MDPs in the full information setting, by reducing to adversarial learning. Reviewers appreciate the greater generality taken by this work, as well as the several improvements and extensions beyond existing literature. On the other hand, the paper does not contain empirical work, which would have offered insights about the technical contributions in reality. Overall, the paper has nice results that are useful and interesting addition to the literature.

While the revised version has made substantial progress in enhancing presentation, the paper still has quite some room for improvement. I’d encourage the authors to take a careful pass and submit a revised version:

* Tighten the language to be less lengthy, remove unnecessary statements, or reword it to elucidate the points you’re making. Eg, “... do not write any explicit name but obtain ...”, “The typical proofs are one-page-long, do not clearly identify a reduction, and consist of ad hoc adaptations of ...”
* Clearer references for easier following. Eg, “Box A in Section 7” is referred to in Section 2 --- maybe consider moving Box A to Section 2, as the reader needs to understand certain parts of it in Sec 2. Another example: it’s unclear what item (ii) refers to in “... handles its term (ii)” (page 5).

**Audience:**

Yes, the work would be interesting to a subcommunities in theoretical machine learning, including adversarial learning and adversarial RL.

**Claims And Evidence:**

The theoretical claims are supported by complete proofs. There are no empirical results.